# Data-mining analysis of the global distribution of soil carbon in observational databases and Earth system models

Shoji Hashimoto[1], Kazuki Nanko[1], Boris Ťupek[2], Aleksi Lehtonen[2]

[1]Forestry and Forest Products Research Institute (FFPRI), Tsukuba, 305-8687, Japan
[2]Natural Resources Institute Finland, Latokartanonkaari 9, Helsinki, FI-00790, Finland

*Correspondence to*: S. Hashimoto (shojih@ffpri.affrc.go.jp)

**Abstract.** Future climate change will dramatically change the carbon balance in the soil, and this change will affect the terrestrial carbon stock and the climate itself. Earth system models (ESMs) are used to understand the current climate and to project future climate conditions, but the soil organic carbon (SOC) stock simulated by ESMs and those of observational databases are not well correlated when the two are compared at fine grid scales. However, the specific key processes and factors, as well as the relationships among these factors that govern the SOC stock, remain unclear; the inclusion of such missing information would improve the agreement between modelled and observational data. In this study, we sought to identify the influential factors that govern global SOC distribution in observational databases, as well as those simulated by ESMs. We used a data-mining (machine-learning) scheme (boosted regression trees: BRT) to identify the factors affecting the SOC stock. We applied BRT to three observational databases and 15 ESM outputs from the fifth phase of the Coupled Model Intercomparison Project (CMIP5) and examined the effects of 13 variables/factors categorized into five groups (climate, soil property, topography, vegetation, and land-use history). Globally, the contributions of mean annual temperature, clay content, CN ratio, wetland ratio, and land cover were high in observational databases, whereas the contributions of the mean annual temperature, land cover, and NPP were predominant in the SOC distribution in ESMs. A comparison of the influential factors at a global scale revealed that the most distinct differences between the SOCs from the observational databases and ESMs were the low clay content and CN ratio contributions and the high NPP contribution in the ESMs. The results of this study will aid in identifying the causes of the current mismatches between observational SOC databases and ESM outputs and improve the modelling of terrestrial carbon dynamics in ESMs. This study also reveals how a data-mining algorithm can be used to assess model outputs.

## 1 Introduction

Soil is the largest organic carbon stock in terrestrial ecosystems (Batjes, 1996; IPCC, 2013; Köchy et al., 2015). The soil organic carbon (SOC) stock represents a balance between carbon inputs to soil and carbon losses from soil via decomposition and dissolved organic carbon, and this influx and efflux of soil carbon are controlled directly and indirectly by environmental conditions (Carvalhais et al., 2014; Schimel et al., 1994). Future climate change will dramatically affect the global soil carbon balance (Bond-Lamberty and Thomson, 2010; Crowther et al., 2016; Friedlingstein et al., 2006; Hashimoto et al., 2011, 2015), and this change will affect terrestrial carbon and consequently the climate itself (Cox et al., 2000; Zaehle, 2013).

Earth system models (ESMs) were developed to understand the current climate and provide future climate projections, and these models incorporate the terrestrial carbon cycle, including SOC (Arora et al., 2013; Friedlingstein et al., 2014). In ecosystem carbon cycle models of ESMs, SOC is calculated as the balance between carbon inputs via dead organic matter and carbon emissions via organic matter decomposition, with both processes influenced by temperature and water conditions. SOC dynamics have a critical influence on the land carbon sink in ESM simulations (Friedlingstein et al., 2014).

Observational global soil databases are often used as benchmarks to examine whether ESMs successfully describe the global distribution of soil carbon stocks (Anav et al., 2013; Hararuk et al., 2014; Todd-Brown et al., 2013; Wieder et al., 2014). Several global soil databases have been developed over the past two decades, and several are undergoing further improvement (Scharlemann et al., 2014). Certain databases describe the global distribution of soil physiochemical properties and enable

calculations of the global distribution of SOC stocks (e.g., Harmonized World Soil Database (HWSD)), whereas others provide SOC stocks by default (e.g., International Geosphere Biosphere Programme's (IGBP) Data and Information System (DIS) database). These databases incorporate observed data points with global coverage, although there are biases in the spatial distribution or density of the data points. In these databases, gridded SOC data have been generated by linking the soil properties to soil maps or by inter-extrapolating the model outputs derived from analyses of observed SOC data points.

Compared with the SOC distribution derived from ESMs, SOC estimates derived from SOC observations are more data oriented; however, even the observational databases include significant uncertainty because of errors in the source data and building processes (Köchy et al., 2015; Todd-Brown et al., 2013).

A recent study (Todd-Brown et al., 2013) found that although ESM results are moderately consistent at the biome level, the correlation between the distribution of soil carbon stocks simulated by ESMs and observational databases is poor when the

two are compared at fine scales (e.g., a 1° scale). Furthermore, estimates of SOC by ESMs and terrestrial biosphere models exhibit high uncertainty (Nishina et al., 2014, 2015; Tian et al., 2015). Several studies have examined the cause of the inconsistency in data derived by observational databases and ESMs and the high variation of SOC outputs from ESMs (Exbrayat et al., 2013; Todd-Brown et al., 2013; Wieder et al., 2013). Todd-Brown et al. (2013) analysed the soil carbon outputs from 11 ESMs from the fifth phase of the Coupled Model Intercomparison Project (CMIP5) and soil carbon data from

the HWSD and found that net primary productivity (NPP) and temperature could explain the SOC spatial variations in the ESM output but not in the HWSD output. These authors also found that the differences in SOC from the ESMs were driven by differences in the simulated NPP and the parameterization of soil heterotrophic respiration and not by differences in the soil model structure of the ESMs. The key influence of parameterizing soil heterotrophic respiration (e.g., turnover time) on SOC in the CMIP5 ESM has also been discussed by Exbrayat et al. (2013). Anav et al. (2013) examined the relationships

between simulated SOC and vegetation carbon and compared them with reference data obtained from observational databases, and they found that although simulated values clustered around the reference values, the ratio of SOC to vegetation carbon differed among the models. This finding suggests that the parameterization of plant production, mortality, and decomposition vary greatly among ESMs. More realistic representations of turnover times (Koven et al., 2015) and focusing on the model treatment of the hydrological cycle on the carbon cycle (Shao et al., 2013) are suggested as future improvements.

Despite these research results, the key processes and factors that govern the SOC stock and the relationships among them remain unclear. The appropriate inclusion of these processes/factors would improve the consistency between the model results and observational data. In this study, we sought to identify the key factors that govern the global SOC distribution in observational databases as well as those simulated by ESMs. We applied a data-mining (machine-learning) scheme (boosted regression tree: BRT) to identify the influential factors and explore how they relate to SOC stocks (Elith et al., 2008). The

BRT method is based on regression trees and boosting. We combined the potentially influential variables from many data products and SOC data from observational databases and ESMs and examined the factors influencing the distribution of SOC and the relationships between these factors and SOC stocks. We assessed how closely ESMs could match the influential factors and their relationships with factors obtained from observational databases. By comparing the influential factors in the observational databases with those in the ESMs, we clarified the model-data discrepancies and the areas in which ESMs can

be improved.

## 2 Materials and methods

### 2.1 Observational global SOC database

We used SOC data from two global and one northern observational database. The first global database was the HWSD (FAO/IIASA/ISRIC/ISSCAS/JRC, 2012). The HWSD is a global database of soil physiochemical properties that has been developed by the International Institute for Applied Systems Analysis (IIASA) and the Food and Agriculture Organization of the United Nations (FAO) in collaboration with the International Soil Reference and Information Centre (ISRIC) -World Soil Information, the European Commission Joint Research Centre (JRC), and the Institute of Soil Science, Chinese Academy of Sciences (ISSCAS). The database was constructed by compiling the European Soil Database (ESDB), a 1:1 million soil map of China, various regional SOTER databases (SOTWIS Database), and a soil map of the world from the FAO. We used an SOC stock database obtained with HWSD from the Joint Research Centre (JRC) (Hiederer and Köchy, 2011) (Fig. 1a). The second database included global gridded surfaces of selected soil characteristics (IGBP-DIS) (Global Soil Data Task Group, 2000) (Fig. 1b), which contains gridded soil physiochemical properties. The database has been developed by the Global Soil Data Task Group of the International Geosphere Biosphere Programme's (IGBP) Data and Information System (DIS), and the database was generated by linking the pedon records in the Global Pedon Database to the FAO/UNESCO digital soil map of the world. The third database was the Northern Circumpolar Soil Carbon Database, version 2 (NCSCD) (Hugelius et al., 2013; Tarnocai et al., 2009) (Fig. 1c). This database is a spatial database of SOC stock of the northern circumpolar permafrost region. The soil map data were obtained from different regions/countries (e.g., USA, Canada, Russia etc.) and were harmonized. The NCSCD were based on 1778 pedon data points.

We used the HWSD and IGBP-DIS to analyse the global distribution of SOC stocks; then, we extracted a database of northern circumpolar regions from the three above databases and analysed the SOC stocks in the northern region. The relationships among the databases are shown in Fig. S1. The SOC in the upper 100 cm in each database was used.

### 2.2 Global SOC estimated using Earth system models

The global distribution of SOC stocks estimated by ESMs was obtained from CMIP5. We examined the results of 15 ESMs (Fig. 2) (Table 2). When more than one result was obtained by the same model family (e.g., MIROC-ESM and MIROC-ESM-CHEM), we generated an ensemble average database for each family (e.g., average of MIROC-ESM and MIROC-ESM-CHEM): Todd-Brown et al. (2013) showed through a hierarchical cluster analysis that SOC distributions were very similar among ESMs from the same climate centre. The mean values from 1980–2004 were calculated. The results of the historical and ensemble member r1i1p1 were used in this study. The notation "r1i1p1" is an identifier of the model simulation and is an ensemble member that is often used for analyses (Chang et al., 2012; Dirmeyer et al., 2013; Jiang et al., 2015; Kumar et al., 2014). The overviews of SOC submodels in the ESMs have been previously described (Exbrayat et al., 2014; Todd-Brown et al., 2013, 2014) and are also shown in Table 2. In general, each soil submodel consisted of 1 to 9 pools and incorporated the effects of temperature and moisture. Some ESMs have litter carbon pools; these were excluded from this study. A comparison between the mean of ESMs and global observational databases in a 1° grid is shown in Fig. S2.

### 2.3 Other databases

We used five groups of variables/factors to examine their effects on global SOC: climate, soil property, topography, vegetation, and land-use history. Detailed data sources for the databases are described in Table 1. The mean annual temperature, and annual precipitation were used as the climate variables, and the clay content, CN ratio, and texture (Appendix Table A1) were used as the soil variables (0–30 cm). The compound topographic index, elevation, slope, and wetland ratio were used as the topographic indices. The CN ratio was calculated by dividing the carbon density by the nitrogen density. The wetland ratio was calculated by dividing the number of wetland grids at 30 seconds by the total grids at 1°. The lake, reservoir, and river

were not quantified as wetlands and were excluded from the total grids. The land cover type (Appendix Table A2) and NPP were adopted as vegetation indices, and the cropland ratio and human appropriation of net primary production percentage, which is a percentage of human consumption of NPP to local NPP (Imhoff and Bounoua, 2006), were used as the indices of land-use history. The average human appropriation of the NPP percentage was calculated at 1°. Histograms of the variables are shown in Fig. S3.

## 2.4 Database handling

All global databases, except for the databases with a spatial resolution of 1° by default, including observational and ESM model outputs, were regridded to a spatial resolution of 1° for the analyses. Regridding of data in the NetCDF format was performed using the Climate Data Operators (CDO) software, version 1.6.9, provided by the Max Plank Institute for Meteorology (https://code.zmaw.de/projects/cdo). A bilinear interpolation, which is one of the most widely used algorithms, was used (remapbil in CDO).

## 2.5 Boosted regression trees (BRT)

To identify the influential factors and their relationships with SOC stocks, BRT were used in this study (Elith et al., 2008). This technique involves a data-mining (machine-learning) algorithm that combines the advantages of a regression tree (decision tree) algorithm and boosting. Regression trees are a classification algorithm that classify data through recursive binary splits, and boosting is a machine-learning algorithm that generates many rough models and combines them to improve their predictive capability. The main advantages of this method are that BRT can analyse different types of variables and interaction effects among variables, and are applicable to nonlinear relationships. In recent years, the BRT technique has been used to examine the distribution of soil characteristics at a regional scale (Aertsen et al., 2011; Cools et al., 2014; Martin et al., 2011). Major outputs from BRT analyses can identify the following: (1) the relative importance (percentage of influence or contribution) of predictor variables (explanatory variables), on the basis of the weighted and scaled number of times a variable is selected for splitting (Elith et al., 2008) and (2) the relationships among variables and the explained variable shown in partial dependence plots.

We used the open-source BRT package (brt.functions.R) in R software version 3.2.1 and 3.2.2 (R Core team, 2013) developed by Elith et al. (2008). The R code for the BRT algorithm is available in the supplementary material of Elith et al. (2008). The gbm package was used (version 2.1.1) to run the BRT package. The calculations were performed in Mac OS X (version 10.9.5 and version 10.10.5). To do so, the "windows" function in the "brt.functions.R" needed to be replaced with the "quartz" function in R. In practice, three parameters in the BRT package— the learning rate ($lr$), tree complexity ($tc$), and bag fraction ($bg$)—control the BRT performance. The $lr$ determines the contribution of each tree, the $tc$ controls the number of splits, and the $bg$ is the proportion of data selected at each step. The number of trees was determined using the cross-validation method in the R package. The maximum number of trees was set to 15,000. The $tc$ value was set to 5. We tested different $lr$ (0.001, 0.005, 0.01, 0.05, 0.1) and $bg$ values (0.5, 0.6, 0.7) and used the best parameter set for each database, but the changes in parameter values had little effect on the model performance.

## 2.6 Model performance

The goodness of fit between the BRT model and data was assessed by using the linear relationship between the predicted and observed values, the coefficient of determination ($R^2$), and the root mean square error (RMSE); it is shown in Tables S2 and S3. For both the observational databases and ESM databases, the BRT models exhibited good performance, with high $R^2$ values in most of the databases, but the performance was relatively lower for NCSCD and CMCC (northern soils).

## 3 Results

### 3.1 Observational databases

#### 3.1.1 Global soil

The relative contributions of variables in the BRT model of global SOC stocks to the observational databases are shown in Fig. 3a and 3b. In HWSD, the contributions of land cover, mean annual temperature, CN ratio, and wetland ratio were high. For IGBP-DIS, the mean annual temperature, followed by clay content, CN ratio, and land cover also highly contributed. In particular, the mean annual temperature was very influential. The contribution of elevation to each HWSD and IGBP-DIS was 6% and 7%, respectively. The NPP contributed 5% in both databases.

The relationships between the influential variables and SOC are shown in Fig. 4a–e. In general, the two databases showed similar relationships. For example, the SOC decreased with increasing mean annual temperature, particularly at sites with a mean annual temperature > 0 °C (Fig. 4a), but increased with increasing clay content and CN ratio (Fig. 4b and 4c). The SOC increased rapidly with an increasing CN ratio. Relationships with the mean annual temperature were similar (Fig. 4a). The relationship with clay was steeper in IGBP-DIS than in HWSD, but the opposite was true for the CN ratio (Fig. 4b and 4c). With respect to land cover, evergreen needleleaf forests and permanent wetlands had higher SOC (Fig. 4e).

#### 3.1.2 Northern soils

In the northern region, the dominant contributors differed among northern soil databases and from those identified in the global database analyses described above (Fig. 3c–e). In HWSD, the CN ratio was the dominant contributor, followed by the wetland ratio, clay content, and mean annual precipitation. In IGBP-DIS, clay content, CN ratio, and elevation were the most important contributors. For NSCD, elevation contributed the most (~25%), but all of the variables except for the cropland ratio and HANPPpct contributed 5–15%. The mean annual temperature was not as influential as the global databases.

The relationships between variables and SOC stock varied more among the databases for northern soils than those of global databases (Fig. 4f–k). Furthermore, because the northern regions were extracted, the ranges of variables were narrower than the global databases. In NCSCD, the SOC decreased with increasing temperature (Fig. 4f) and increased with increasing precipitation (Fig. 4g). The SOC increased with increasing clay content and CN ratio in HWSD and IGBP-DIS (Fig. 4h and 4i), which was consistent with the findings obtained from the global databases. The increasing trend with increasing CN ratio was also observed in NCSCD. The SOC decreased with increasing elevation in all databases but showed considerable variability at low elevations (Fig. 4j).

### 3.2 Earth system models

#### 3.2.1 Global soil

The contributions of some variables varied among ESMs, but the mean of the results of the ESMs showed that the mean annual temperature, land cover, and NPP clearly contributed to SOC distribution (Figs. 5a and 5b). Large inconsistencies between the observational databases and ESMs were found in the low contributions of clay content and the CN ratio and in the high contributions of NPP in ESMs (Figs. 5a and 5b). The contribution of NPP to ESMs was greater than in the observational databases.

The relationships between SOC and certain variables substantially varied among the ESM databases (Fig. 6a–e), particularly in the mean annual temperature (Fig. 6a). The SOC decreased with increasing mean annual temperature (Fig. 6a) but increased with increasing precipitation (Fig. 6b) and NPP (Fig. 6e). The mean of the relationship with mean annual temperature for ESMs was highly consistent with that in the HWSD and IGBP-DIS databases of the temperature range −5–15 °C (Fig. 6a). The increasing trend with increasing NPP in ESMs was consistent with that of the HWSD, particularly below approximately

500 g C m$^{-2}$ of NPP (Fig. 6e). Although the wetland ratio did not contribute to the ESMs (Fig. 6a) with respect to land cover, permanent wetlands had higher SOC (Fig. 6d).

### 3.2.2 Northern soils

The mean of the ESMs showed that for northern soils, the main contributors (mean annual temperature, land cover, and NPP) were mainly the same as in the ESM global outputs (Fig. 5c and Fig. 5d). The contribution of the mean annual temperature was lower than that of the global results of the ESMs (mean of 14% for the northern and 29% for the global temperatures). The relatively large discrepancy between the observational databases and ESMs included the lower contribution of clay content, CN ratio, and elevation and the higher contribution of the mean annual temperature, land cover, and NPP in the ESMs.

The relationship between SOC and variables in ESMs as well as the results of the observational databases are shown in Fig. 6f–i. The mean of the ESMs indicated that the SOC in the northern region increased with increasing NPP, and the relationship was similar to that in HWSD (Fig. 6i), although the contribution of NPP in the ESMs differed from those of the observational database (Fig. 5c). The decreasing trend with elevation was not replicated in the ESMs (Fig. 6g).

## 4 Discussion and concluding remarks

**4.1 Identified influential factors**

Compared with previous studies, we examined the contributions of a wider variety of factors to SOC distributions. Our analyses revealed that the most distinct differences between the observational database data and the ESM outputs were the effects of the CN ratio and clay content (Fig. 5). For both global observational databases, the CN ratio was a substantial contributor (Figs. 3a and 3b). The important contribution of the CN ratio was the same in the northern databases (Fig. 3c–e). The SOC in the

observational databases increased with increases in the CN ratio (Fig. 4c), whereas the SOC values of the ESMs were insensitive to the CN ratio. Our results support the importance of properly incorporating the N cycle into SOC models (e.g., control over decomposition, soil fertility, nutrient availability, and plant litter quality) (Berg et al., 2001; Cotrufo et al., 2013; Fernández-Martínez et al., 2014; Liski et al., 2005; Tuomi et al., 2009; Ťupek et al., 2016). None of the ESMs except for the CESM1 and NorESM in CMIP5 included terrestrial nitrogen processes (Todd-Brown et al., 2013); however, including this

parameter has been suggested as a key improvement for the next model intercomparison (CMIP6) (Hajima et al., 2014; Zaehle et al., 2015). The results of our analysis support the importance of including the N cycle in ESM models.

Clay content is also often used as a regulator of the decomposability of organic matter in the soil (e.g., CENTURY and RothC) (Coleman and Jenkinson, 1999; Parton et al., 1987). Generally, high clay content inhibits organic matter decomposition in the soil. Furthermore, high clay contents often result in low drainage and anaerobic soil conditions, which also inhibit organic

matter decomposition. For the IGBP-DIS data, the contribution of the clay content was as high as that of the CN ratio. The control of decomposability by the clay content has been previously incorporated in site-scale process-based models (Parton et al., 1987) and may be incorporated in certain ESMs because the soil carbon submodels in these ESMs are based on the CENTURY model (see the soil model history reported in Todd-Brown et al., 2014). However, regardless of whether the control of decomposability by clay is incorporated, our results suggest that the influence of clay on the carbon cycle is not well captured

in most ESMs.

The mean annual temperature was identified as an influential factor in the global databases (Fig. 3a and 3b) but not in the northern soil databases (Fig. 3c–e). Temperature is a main factor controlling both plant production (source of carbon input to soil) and soil organic matter decomposition, which are already incorporated in ESMs. Based on an analysis of the output of heterotrophic respiration, the temperature sensitivity (e.g., $Q_{10}$ value) of soil organic matter decomposition in the ESMs has

been reported as 1.4 to 2.2 (Todd-Brown et al., 2014). In addition, our analyses identified diverse relationships between the

mean annual temperature and SOC. The lower contribution of the mean annual temperature in the northern soils likely occurred because temperature sensitivity is an exponential process, and the magnitude of observed changes under changing temperature is relatively small at a low temperature range, as shown in a comparison of temperature functions in common biogeochemical models (Sierra et al., 2015). The relationships between the SOC and temperature obtained in this study include the integration

of the temperature sensitivity of both plant production and soil organic decomposition and thus do not provide the temperature sensitivity parameter of individual processes for ESMs. However, the results of this study can be used to examine the consistency between the ESM outputs and observational databases.

The mean annual precipitation made a moderate contribution to the global observational databases and ESM outputs, which was likely because NPP and temperature were strongly correlated with moisture and the temperature sensitivity of

decomposition is generally more dominant than the soil moisture sensitivity. Similar ESM outputs have been reported in Todd-Brown et al. (2013). However, precipitation does not necessarily represent the actual moisture conditions in soil, and soil moisture conditions are related to climate conditions as well as soil texture, terrain, and vegetation. In addition, aerobic/anaerobic conditions are important because the wetland ratio has been identified as one of the influential factors (described below).

In the ESMs, NPP was selected as an influential factor in analyses of the global and northern SOC (Fig. 5), whereas in the observational databases (Fig. 3), NPP was not an influential factor, which is consistent with the findings of a previous study (Todd-Brown et al., 2013). Todd-Brown et al. (2013) found that one of the major causes of variations in SOC among ESMs is differences in simulated NPP and the strong control by NPP are not observed in the HWSD output. This high NPP contribution in ESMs is understandable because the terrestrial carbon balance is modelled by calculating the SOC stock via NPP or plant

litter input to soil and soil organic matter decomposition because plant litter input is proportionate to NPP. However, our analyses suggest that the influence of NPP on soil organic matter in the observational soil databases was obscured by other factors. When ESMs incorporate the effects of other factors, such as the N cycle, the effect of NPP may be diluted. Moreover, the large variations in the total amount of SOC from ESMs are partly caused by variations in the modelled NPP in each ESM (Todd-Brown et al., 2013). Furthermore, SOC storage occurs via organic matter accumulation over decades and even millennia.

Thus, prior NPP, land fires, and land-use changes may still affect current SOC values (Carvalhais et al., 2008; Wutzler and Reichstein, 2007). Land cover is another important factor, and incorporating the hydrology and resulting carbon dynamics in wetlands may lead to important improvements in ESMs.

Elevation was another influential factor, particularly in the northern observational databases (Fig. 3d and 3e). We speculate that elevation may serve as a comprehensive index of SOC in limited areas because other variables, such as temperature, NPP,

soil texture and other factors, change with increases in elevation. The effect of elevation in ESMs was not as high as that in the observational databases (Fig. 5). We estimated that the effect of elevation might automatically increase if the other aforementioned processes are properly adjusted/included in the ESMs.

## 4.2 Similarity in influential factors for ESMs outputs

Analyses of the ESM outputs showed large variability, although the influential factors were largely similar among the ESMs

(Fig. 5). This similarity likely indicates that the structure of the models that describe SOC dynamics in the ESMs is similar. One reason for the similarity may be that certain ESMs share a common code (Alexander and Easterbrook, 2015). Another reason may be rooted in the basic structure of the soil carbon model, wherein SOC is calculated as the balance between dead organic matter input to soil and carbon emissions from the decomposition of organic matter. These processes are influenced by temperature and water conditions. The SOC pool is characterized by its turnover time (decomposition constant). In general,

decomposition exhibits an exponential response to temperature, which is more severe than its response to water. As a result, model results for SOC are strongly influenced by NPP (litter input), temperature, and turnover time, which have been demonstrated by previous studies (Exbrayat et al., 2014; Todd-Brown et al., 2013) and were confirmed in our analyses.

As shown in Table 2, the submodels of SOCs in ESMs differ in the number of SOC pools and temperature and moisture functions. Todd-Brown et al. (2013) reported that the ESM outputs and observational SOC database results did not produce consistent patterns for soil carbon pools, temperature and moisture sensitivity functions. Exbrayat et al. (2014) found that the turnover times of SOC in the ESM outputs were not affected by the number of SOC pools. Our analyses also indicated that a match or mismatch of major contributing factors between the ESM outputs and observational database results are not strongly related to these properties of SOC submodels. Thus, the spatial pattern of SOC from ESMs are likely more strongly affected by the basic structure, driving variables (NPP and temperature), and parameterizations (turnover time and temperature and moisture sensitivity, which are influential) than by the number of pools and the types of temperature and moisture sensitivity functions. From a mathematical perspective, the similarity is likely fundamentally based on the description of these SOC dynamics by a series of first-order linear ordinary differential equations that are not autonomous (Manzoni and Porporato, 2009; Sierra and Müller, 2015). With these equations, the outputs generally do not show chaotic behaviours.

## 4.3 Uncertainty and other factors

Observational databases are directly generated from SOC observations; therefore, these databases should be closer to the real SOC distribution than databases based on ESM outputs. Hence, observational databases are often used as benchmarks to evaluate the outputs of ESMs. Still, we should be aware that these observational databases are generated using assumptions, certain algorithms, and uncertain inputs. In particular, the uncertainty in the SOC for the northern regions is high in the observational databases (Fig. 1), which was also observed in the results from our BRT analysis (Fig. 3 c–e). When the estimated SOC distribution from various approaches, such as data-driven and process-oriented modelling, are consistent, then our estimations have high confidence.

Although we examined key factors from a wide variety of candidate properties, potentially important mechanisms that would improve the reproducibility of SOC distributions by ESMs and process-based ecosystem models may still be missing. For example, including microbial dynamics in SOC models may improve projections of global soil carbon by ESMs (Wieder et al., 2013), although models that include these dynamics are still in development (see Wang et al., 2014, 2016; Wieder et al., 2015). Reports indicate that the role of mycorrhizae in soil carbon storage is important (Averill et al., 2014). Because soil carbon accumulation and decomposition are slow processes and land cover is an important factor in SOC, as shown in our study, considering the land-use history of an area may be essential for improving ESMs. Furthermore, because soil has depth and SOC and soil environments vary according to depth (Davidson and Trumbore, 1995; Hashimoto and Komatsu, 2006; Jobbágy and Jackson, 2000), vertical soil heterogeneity/processes are important (Braakhekke et al., 2013; Wieder et al., 2013). The importance of mineral reactivity has also been suggested (Doetterl et al., 2015). However, our results suggest that the performance of ESMs can be improved simply through adequate re-evaluation/inclusion of well-known processes. Another approach for improving ESMs is model-data fusion (assimilation) (Hararuk et al., 2014). Although its application to a part of an ESM (e.g., ecosystem carbon cycle model) is realistic in consideration of the long running time, constraining model parameters with observational databases via data assimilation, such as a Bayesian approach, would improve the performance of ESMs. Another uncertainty of this analysis is the issue of scale because analyses applied at much finer resolutions, such as 1 km, might be governed by different influential factors. The potential mechanisms, parameterization, and other modelling issues for next-generation ESMs are not limited to those listed above and have been thoroughly discussed elsewhere (Luo et al., 2016; Ostle et al., 2009; Wieder et al., 2015).

## 4.4 Concluding remarks

Although observational estimations of SOC are still under development and have significant uncertainty, the consistency between observational SOC database results and ESM outputs will enhance our confidence in predicting SOC dynamics under climate change. In this study, the same data-mining BRT algorithm was applied to observational databases of SOC stocks and

ESM outputs. By comparing the outputs from both analyses, we revealed the similarities and differences among the observational databases and ESMs. On a global scale, in addition to improving the parameterization of temperature sensitivity and NPP, properly incorporating the influence of the nitrogen cycle and clay content in ESMs was identified as a potential method of improving the ability of these models to reproduce the distribution of SOC found in observational databases. The

results of this study should help to identify the causes of the current mismatches between observational SOC databases and ESM outputs and improve the terrestrial carbon dynamics modelled in ESMs. This study demonstrates that the data-mining scheme can be used to compare the results from observational databases and ESMs in detail and determine the key factors involved in the mismatches.

**Code and Data availability**

The R code, with a tutorial, for the BRT algorithm is available in the supplementary material of Elith et al. (2008) (http://onlinelibrary.wiley.com/doi/10.1111/j.1365-2656.2008.01390.x/full). The codes and data for the observational databases are available in the Supplement.

**Acknowledgements**

This study was supported by JSPS KAKENHI Grant Number 24510025. We also acknowledge the Academy of Finland and

mobility funding (nr. 276300) for supporting this work. The constructive comments by Ben Bond-Lamberty and Katherine Todd-Brown as reviewers greatly improved the manuscript. We thank Dr. Tomohiro Hajima for helping to improve our understanding of the CMIP5 models, and we also thank Dr. Atsushi Obata for information about the MRI ESM model. We acknowledge the World Climate Research Programme's Working Group on Coupled Modelling, which is responsible for CMIP, and we thank the climate modelling groups (listed in Table 2 of this paper) for producing and making their model

output available. For CMIP, the U.S. Department of Energy's Program for Climate Model Diagnosis and Intercomparison provided coordinating support and led development of software infrastructure in partnership with the Global Organization for Earth System Science Portals.

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

**Table**

Table 1. Variables used in the analyses and their sources.

| Variable | Abbreviation | Source (database) | Original resolution | Reference |
|---|---|---|---|---|
| Mean annual temperature[1] | MAT | ISLSCPII (CRU05) | 1 ° | New et al., 2011 |
| Mean annual precipitation[1] | MAP | ISLSCPII (CRU05) | 1 ° | New et al., 2011 |
| Clay content (0–30 cm) | Clay | ISLSCPII | 1 ° | Scholes and Brown de Colstoun, 2011 |
| CN ratio (0–30 cm)[2] | CN ratio | ISLSCPII | 1 ° | Scholes and Brown de Colstoun, 2011 |
| Soil texture (0–30 cm) | Texture | ISLSCPII | 1 ° | Scholes and Brown de Colstoun, 2011 |
| Compound topographic index[3] | CTI | ISLSCPII | 1 ° | Verdin, 2011 |
| Elevation[3] | Elev. | ISLSCPII | 1 ° | Verdin, 2011 |
| Slope[3] | Slope | ISLSCPII | 1 ° | Verdin, 2011 |
| Wetland ratio | Wetland | Global Lakes and Wetlands Database | 30 sec | Lehner and Döll, 2004 |
| Land cover | LandCover | ISLSCPII | 1 ° | Friedl et al., 2010 |
| Net primary production | NPP | ISLSCPII | 1 ° | Prince and Zheng, 2011 |
| Cropland ratio | Cropland | ISLSCPII | 1 ° | Ramankutty and Foley, 2010 |
| Human appropriation of NPP percentage | HANPPpct | HANPP collection | 0.25° | Imhoff et al., 2004 |

[1] The original database provides monthly data. Annual means were calculated by the authors.

[2] The CN ratio was calculated by dividing the carbon density by the nitrogen density.

[3] The native database is hydro1k, and its resolution is 1 km. The mean value of 1 km was used in this study.

Table 2: ESMs used as outputs in this study. The term "ensemble" indicates the ensemble of outputs from the same families. The number of soil pools, types temperature sensitivity function, types of moisture sensitivity function, and link to nitrogen cycling. URLs of model/modelling group/model description paper are also shown. Model names and modelling centres are shown in Table S1.

| ESM[*1] | Number of Pool[*2] | Temperature sensitivity[*2] | Moisture[*2] | Nitrogen[*2] | URL of model or modelling group or model description paper |
|---|---|---|---|---|---|
| BCC-ensemble | 6 | Hill | Hill | No | http://forecast.bcccsm.cma.gov.cn/htm/ |
| BNU-ESM | 2 | Arrhenius | Increasing | No | http://esg.bnu.edu.cn/BNU_ESM_webs/htmls/index.html |
| CanESM2 | 1 | $Q_{10}$ | Hill | No | http://ec.gc.ca/ccmac-cccma/default.asp?lang=En&n=4596B3A2-1 |
| CCSM4 | 3 | Arrhenius | Increasing | Yes | http://www.cesm.ucar.edu/models/ccsm4.0/ |
| CESM1-ensemble | 3 | Arrhenius | Increasing | Yes | http://www.cesm.ucar.edu/models/cesm1.0/ |
| CMCC-CESM | 3[*3] | Unknown | Unknown | Unknown | http://www.cmcc.it/models/cmcc-esm-earth-system-model |
| GFDL-ESM2M | 2 | Hill | Increasing | No | https://www.gfdl.noaa.gov/earth-system-model/ |
| GISS-ensemble | 9 | Increasing | Increasing | No | http://www.giss.nasa.gov/tools/modelE/ |
| HadGEM2-CC | 4 | $Q_{10}$ | Hill | No | http://www.metoffice.gov.uk/research/modelling-systems/unified-model/climate-models/hadgem2 |
| INM-CM4 | 1 | $Q_{10}$ | Hill | No | - |
| IPSL-ensemble | 4 | $Q_{10}$ | Increasing | No | http://icmc.ipsl.fr/index.php/icmc-models |
| MIROC-ensemble | 2 | Arrhenius | Increasing | No | http://www.geosci-model-dev.net/4/845/2011/ |
| MPI-ensemble | 3 | $Q_{10}$ | Increasing | No | http://www.mpimet.mpg.de/en/science/models/mpi-esm.html |
| MRI-ESM1 | 2[*3] | Arrhenius[*4] | Increasing[*4] | No[*4] | http://www.mri-jma.go.jp/Publish/Technical/DATA/VOL_64/index_en.html |
| NorESM1-ensemble | 3 | Arrhenius | Increasing | Yes | https://wiki.met.no/noresm/start |

[*1]BCC-ensemble: BCC-CSM1.1 and BCC-CSM1.1(m), CESM1-emsemble: CESM1(BGC), CESM1(CAM5), CESM1(FASTCHEM), and CESM1(WACCM), GISS-ensemble: GISS-E2-H, GISS-E2-H-CC, GISS-E2-R, and GISS-E2-R-CC, IPSL-ensemble: IPSL-CM5A-LR, IPSL-CM5A-MR, and IPSL-CM5B-LR, MIROC-ensemble: MIROC-ESM and MIROC-ESM-CHEM, MPI-ensemble: MPI-ESM-MR and MPI-ESM-LR, and NorESM1-ensemble: NorESM1-M and

10   NorESM1-ME. [*2]Adopted from Todd-Brown et al. 2013, 2014. [*3]Technical reports. [*4]Personal communications.

Appendix

Table A1: Classification of soil texture in ISLSCPII (see Table 1).

| ID | Texture |
|----|---------|
| 1 | Sand |
| 2 | Loamy Sand |
| 3 | Sandy Loam |
| 4 | Silt Loam |
| 5 | Silt |
| 6 | Loam |
| 7 | Sandy Clay Loam |
| 8 | Silt Clay Loam |
| 9 | Clay Loam |
| 10 | Sandy Clay |
| 11 | Silty Clay |
| 12 | Clay |

Table A2: Classification of land cover in ISLSCPII (see Table 1).

| ID | Land cover |
|----|------------|
| 1 | Evergreen Needleleaf Forest |
| 2 | Evergreen Broadleaf Forests |
| 3 | Deciduous Needleleaf Forests |
| 4 | Deciduous Broadleaf Forests |
| 5 | Mixed Forests |
| 6 | Closed Shrublands |
| 7 | Open Shrublands |
| 8 | Woody Savannahs |
| 9 | Savannahs |
| 10 | Grasslands |
| 11 | Permanent Wetlands |
| 12 | Croplands |
| 13 | Urban and Built-Up |
| 14 | Cropland/Natural Vegetation Mosaic |
| 15 | Permanent Snow and Ice |
| 16 | Barren or Sparsely Vegetated |

**Figures**

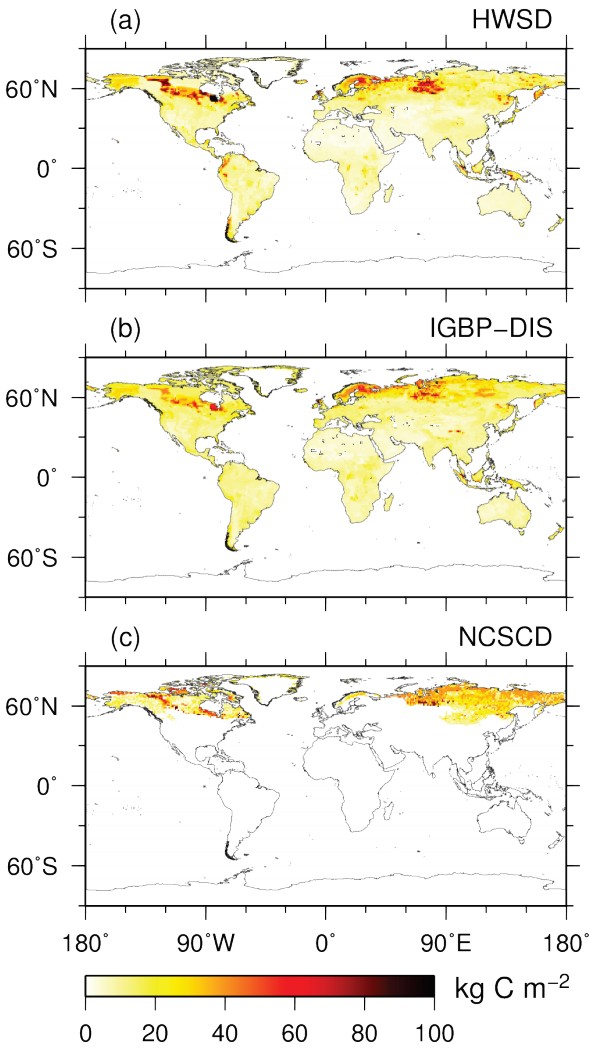

Figure 1. Soil carbon stock in the upper 100 cm (kg C m$^{-2}$) from the observational databases (HWSD, IGBP-DIS, and NCSCD).

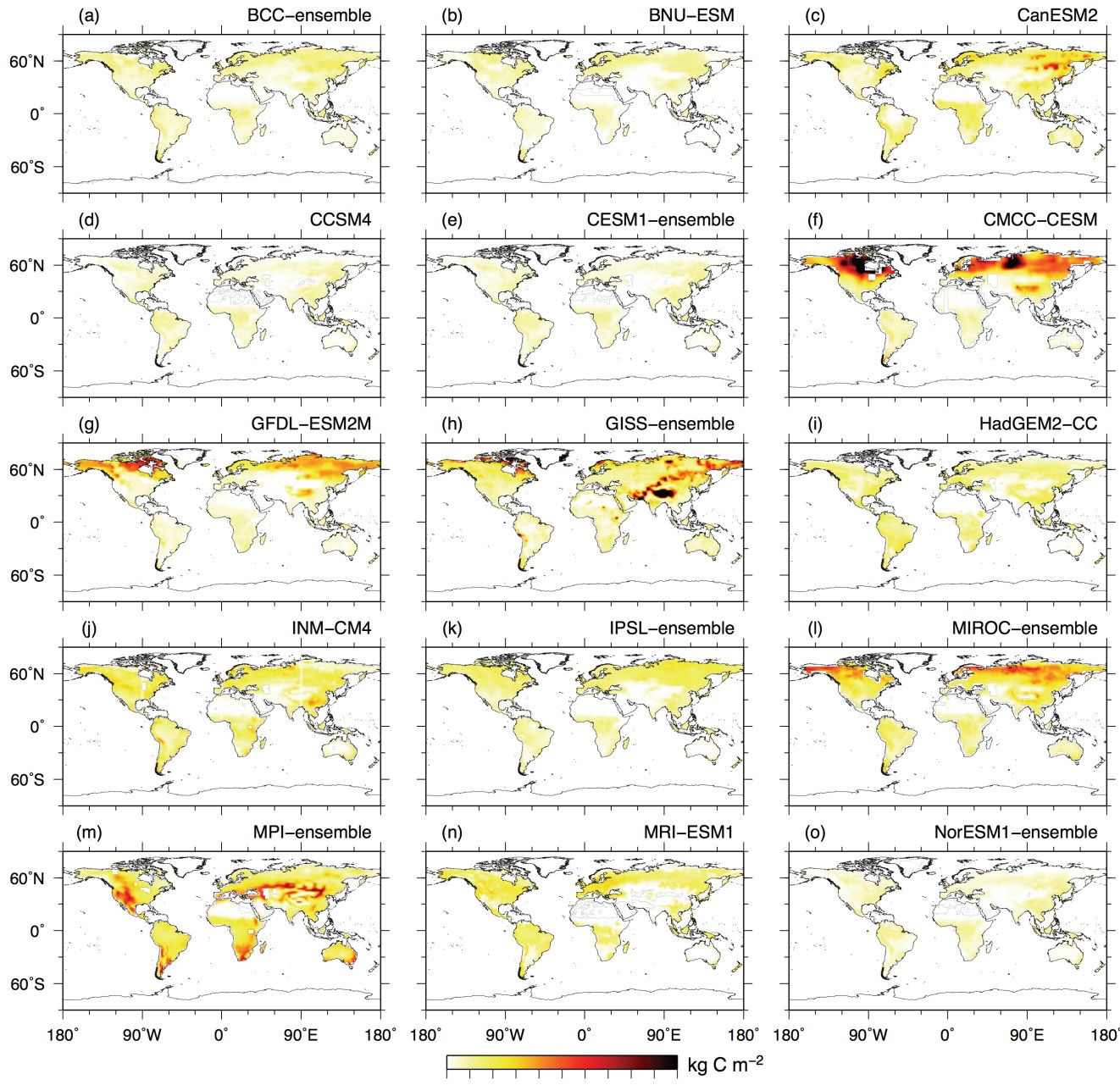

Figure 2. Soil carbon stocks (kg C m$^{-2}$) from Earth system models (CMIP5). The term "ensemble" indicates the result of an ensemble of family members.

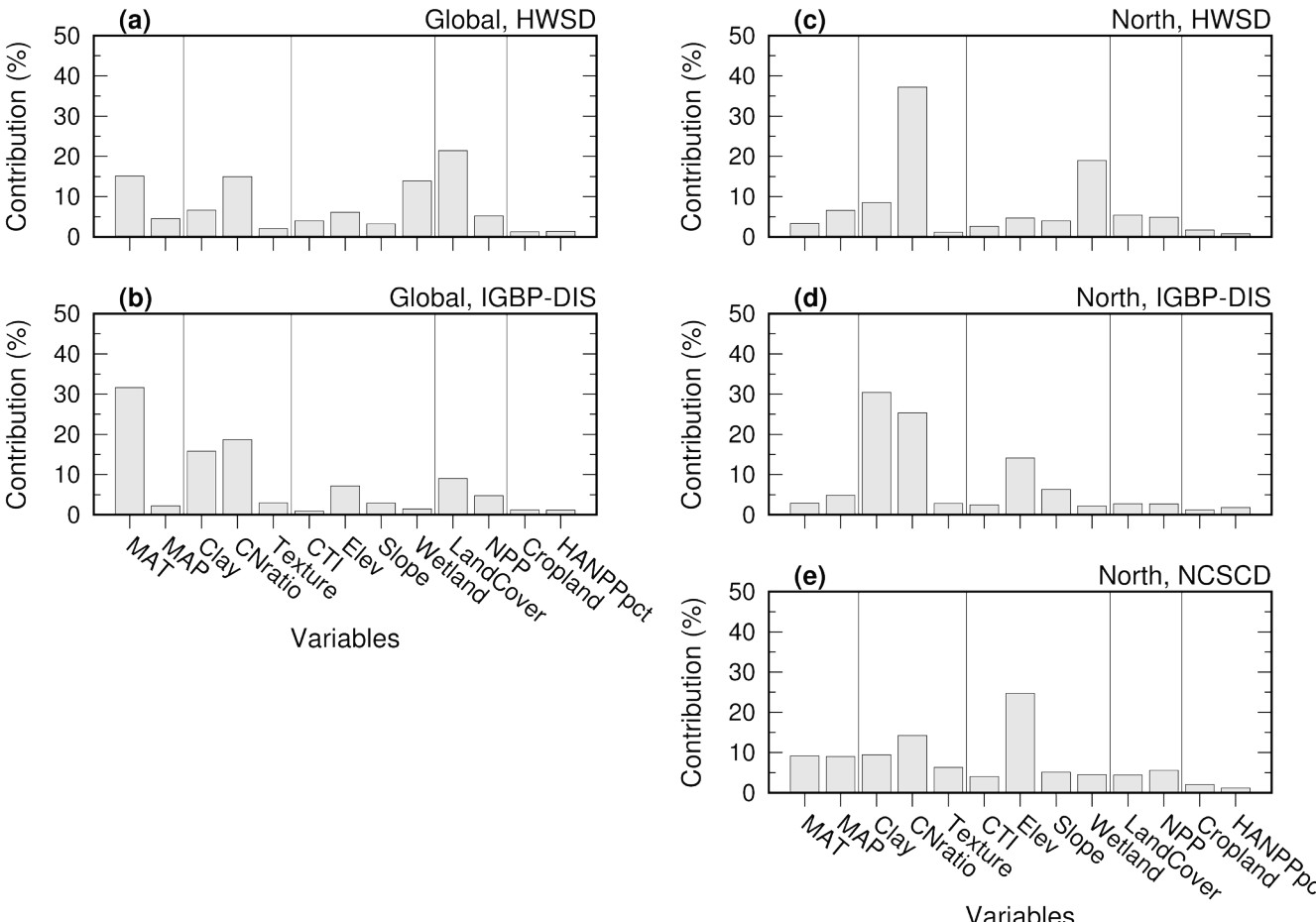

Figure 3. Relative contribution (influence) of predictive variables for the model of soil carbon stocks in the global observational databases (left) and northern observational databases (right).

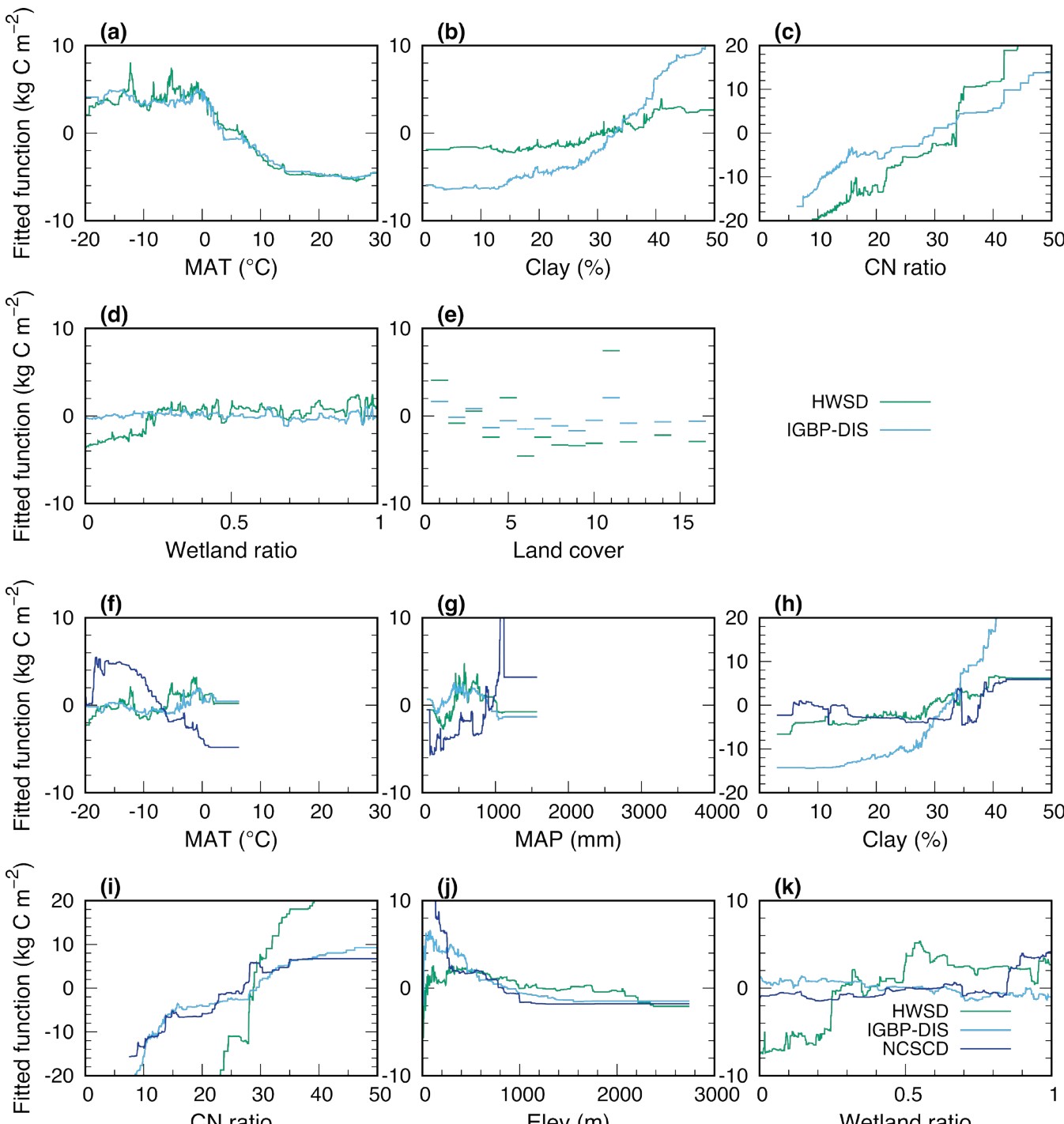

Figure 4. Effects of the most influential variables in the model of the soil carbon stock for each global (a–e) and northern (f–k) observational databases. The fitted functions were centred by subtracting their means. See Table A2 for land cover classifications. Because of the small number of data points, the results for "15, Permanent Snow and Ice" are not shown (e). The y-axis scales for clay and the CN ratio are different from those of other factors (c, h, i).

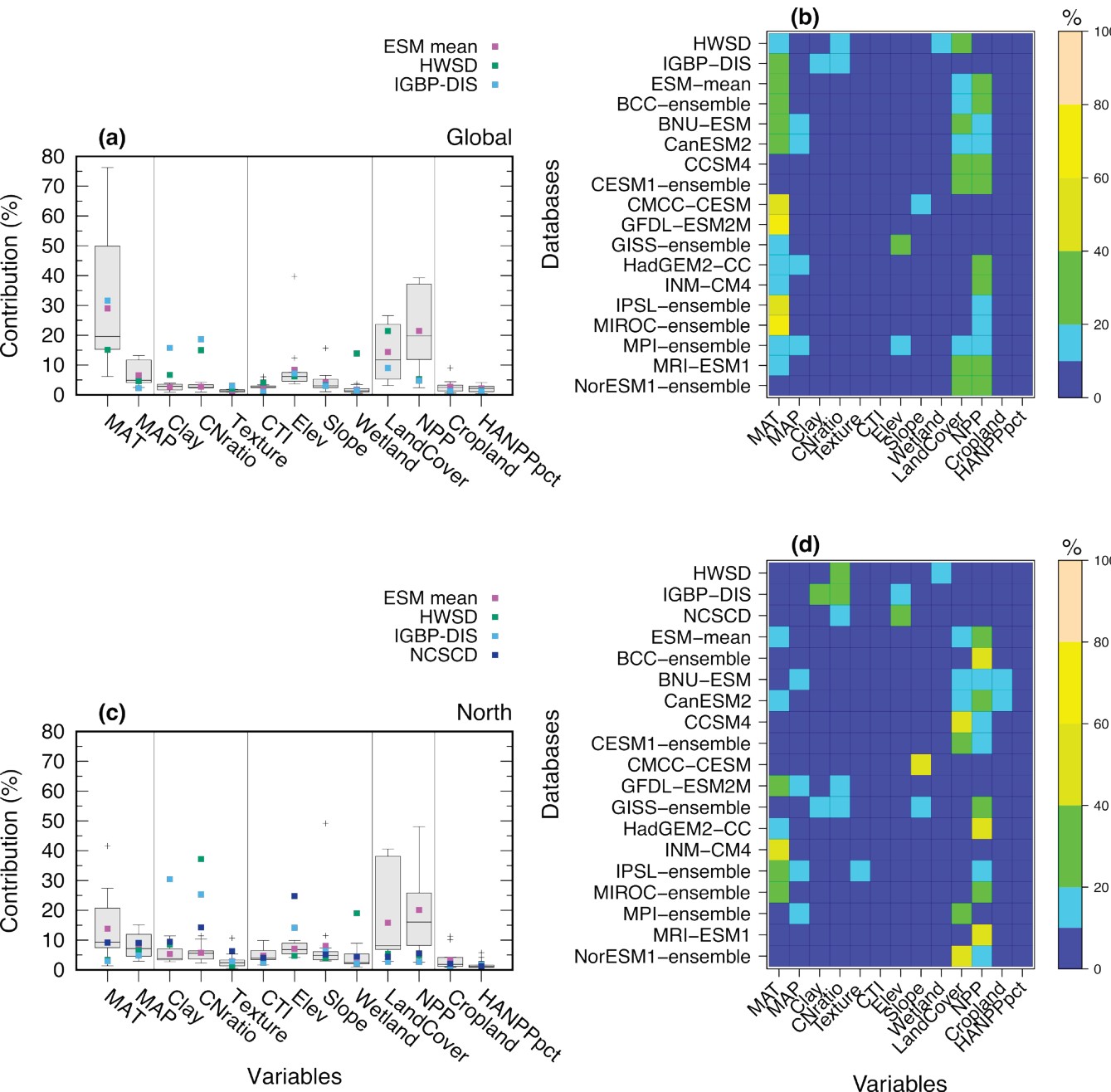

Figure 5. Relative contribution (influence) of predictive variables for the model of the soil carbon stock from ESMs and a comparison with those of observational databases. Box plots show the results of ESM, and the purple, green, light blue, and blue marks indicate the mean of the ESMs and results from observational databases (a: global; c: north). Mosaic plots of detailed relative contributions for each ESM (b: global; d: north).

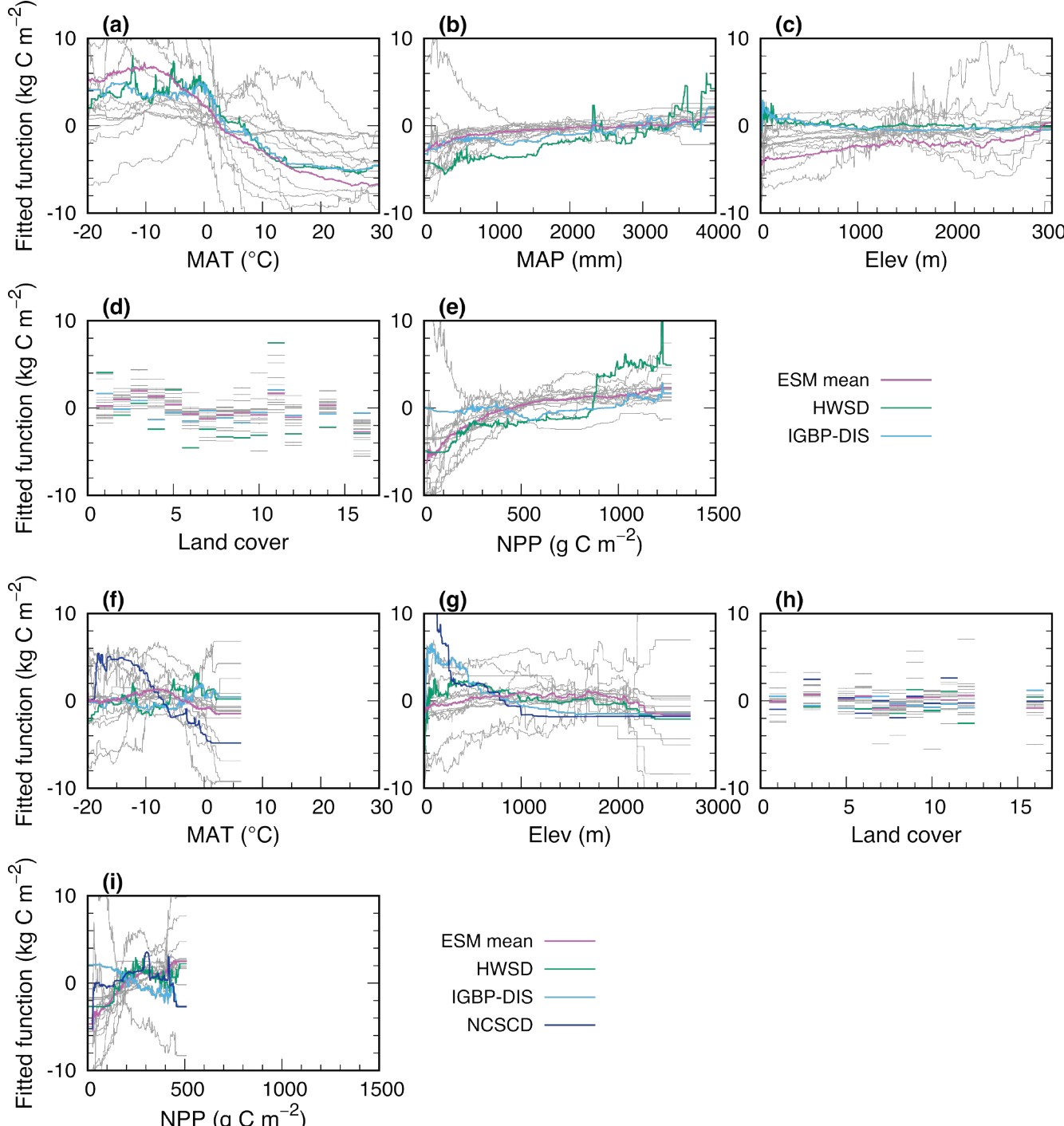

Figure 6. Effect of the most influential variables in the model for global (a–e) and northern (f–i) outputs from ESMs and a comparison with those of observational databases. Grey lines show the results of each ESM, and the purple line indicates the mean of the ESMs. The fitted functions were centred by subtracting their means. See Table A2 for land cover classifications. Because of the small number of data points, the results for "15, Permanent Snow and Ice" are not shown (d, h).