# Peer review of "Data-mining analysis of the global distribution of soil carbon in observational databases and Earth system models"

_Geoscientific Model Development, 2016_

## Referee Comment (RC1) · Anonymous Referee #1 · 3 Aug 2016

General comments

===============

This manuscript describes an analysis, using machine-learning algorithms, of what factors affect the distribution of soil organic carbon (SOC) in both observational databases (e.g., the Harmonized World Soil Database) and earth system models (ESMs, in particular CMIP5 data). This is an important and interesting topic, as our understanding of the factors governing the spatial distribution of SOC is poor. Data-driven algorithms such as those used here offer the possibility of novel, quantifiable insights into both natural processes and model weaknesses.

[Figure]

**GMDD**

This ms is thus promising, but ultimately significantly weakened by a series of problems. First, poor presentation: many parts of the text are unclear; some of the figures need re-thinking; results are at times presented confusingly.

Second, there's no reproducibility, which is shocking (see #6 below). In particular, no code or data availability is specified, no software details given, nor are the methods fully complete or understandable.

Finally, and related to the previous point, there's a lack of insight and applicability. Most of the discussion is a rote recitation of formulaic points; the authors need to expand on the genuinely interesting parts, and offer more interesting, novel insights on how their results apply, and will be useful, to future work. The lack of reproducibility (above) means that it's also not clear how any of this would inform or be useful for modelers seeking to improve their software and science.

In summary, there are interesting points here, but the current ms needs substantial revisions in almost every area for clarity and presentation, reproducibility, and insight.

Specific comments

===============

1. Page 1, line 1: I'd suggest either "Data-mining analysis of the global..." or "Factors affecting the global..."

2. P. 1, lines 9, 20, and 26-27: these three short sentences could be deleted with no real loss

3. P. 1, l. 25: "elucidate the nature" of the databases? Confusing

4. P. 2, l. 4: what recent study?

5. P. 3, l. 8-9: divided over what spatial scale? Some more detail in this entire paragraph would be useful

6. Methods: need to give version numbers CDO, R, and all packages used. Also, I'm shocked at the complete lack of any mention of data or code availability (no, that one sentence on p. 7 doesn't count). It's 2016, and I expect all code and data (at least that backing the main results) to be included as supplementary info, or posted in a repository. It's not acceptable to produce results from a black box; see also http://www.geoscientific-model-development.net/about/code_and_data_policy.html

7. P. 4, l. 17: "Relationships with a mean annual temperature were relatively close to each other" – what does this mean? Clarify

8. P. 4, l. 34: "The contribution of each variable varied between ESMs" ?

9. P. 4, l. 36: "large inconsistencies...demonstrated low contributions" – what?

10. P. 5, l. 23-24: this is an interesting point, and should be expanded upon. What are the implications, if the seemingly wide variety of CMIP5 models in fact uses a much smaller number of fundamental assumptions or modeling approaches? I'm pretty sure that Kathe Todd-Brown made this point in one of her papers; see also Alexander et al. (2015), 10.5194/gmd-8-1221-2015

11. P. 6, l. 13-14: "The use of temperature sensitivity..." - ?

12. P. 7, l. 2-3: would such model-data fusion ever be possible, given the extremely long running time of modern ESMs?

13. Table A1: an URL or reference for each model would be useful

14. Table A2: this classification was applied to...? Where is it from?

15. Figures 1 and 2: these are so tiny I'm not sure they convey any information, really

16. Figure 6 should be the central, most important figure of the entire paper–showing how variable importance compares between observational databases and ESMs–but it's very difficult to see what's going on. I'd suggest re-thinking this, and carefully considering the most effective way to show this

---

## Referee Comment (RC2) · K. Todd-Brown (Referee) · 17 Aug 2016

This study used boosted regression tress to identify mismatches in drivers of soil carbon data products and simulated soil carbon in Earth system models. Given how complex Earth system models have become studies like this one which apply statistical methods to both data products and simulation results can provide key insights into what is driving model behavior and contrast that with data driven statistical models. Resulting in better targeted model development.

While I appreciated the authors attempt at brevity I believe that they went a little too far, leaving out key details which are needed to reproduce the results and falling just short of relevant interpretation. The authors need a more detailed overview of the model

structure of the ESMs examined in this study and whether or not that structure played a role in the attribution. Do you see any effects of model structure on the BRT results? Can these results be tied to particular temperature sensitivity function or number of soil carbon pools represented in the simulation?

Finally this study needs to be placed in context of other studies which have examined the different driving variables in ESMs, particularly the CMIP5 models. While the authors offer some token mentions there is a painful lack of detail on this topic.

Please see below for a line by line reaction:

Please make it clearer that the ESMs are regressed against data products not other ESM output. While the modern ESM NPP and temperature distributions match better with current data products. There are some notable differences in modeled NPP in particular in the CMIP5 models and this could be a source of bias in the analysis.

P1L23-24 C:N ratio and clay content are in most ESMs in the allocation scheme. While it is intractable to investigate each modeling code directly Much of the documentation for these ESMs includes Lignin:N ratio (similar to C:N ratios) and clay content mediating decomposition. CENTURY Parton et al 1988 use Lignin:N ratio and clay content for allocation parameters (IPSL-CM5 Krinner etal 2005 cite CENTURY: Parton et al 1988)

P2L4-8 Should there be a citation here?

P2 L8-11 A more in depth treatment of past attempts to disentangle drivers of data-model differences is called for here. Please expand on each of these treatments with particular attention to the ones that looked at the same models and data products the authors are using in this study. In addition, add something to the discussion to contrast your results with these studies.

Section 2.1 There needs to be some discussion about model structure in the ESM vs data products. These data products are typically constructed using correlation to the local environment (climate + land cover + geology) where the pedon was collected.

[Figure]

Please summarize the methods used for each specific data product. For ESMs a discussion of their sensitivity functions and pool structure is appropriate (note that BCC was incorrectly stated to have their N-cycle turned on for CMIP5 in Todd-Brown et al 2013).

P2 L33-35 Be more convincing about averaging models from the same center, there is some clustering analysis that is in the supplemental of Todd-Brown et al 2013 that could support this.

P2 L34-35 Todd-Brown et al 2013 averaged all ensembles that were available at the time, this statement is incorrect. Please either provide a different justification for only considering one ensemble or, preferably, go back and re-analyze the data with the multi-ensemble mean (even better if you can incorporate the modeled uncertainty).

Section 2.4 What regridding algorithm did you use? There are several options in CDO, not all are appropriate for soil data, temperature and NPP. Please discussion which algorithm was used and why.

P5 L1 Describe the results here in addition to referencing the figure.

P7 L15 A BRT tutorial is not appropriate to cite under 'Code availability'. Please either link or reference as SI to the actual code used in this analysis (preferred) or remove this section.

---

## Author Comment (AC1) · 7 Oct 2016

Referee #1

We greatly appreciate your thoughtful and constructive comments. We have revised the manuscript on the basis of your comments, and our responses to the Major and Specific comments can be found below. According to the editorial instructions, our response is structured as follows: (1) comments from Referees; (2) author's response; and (3) author's changes to the manuscript. Thank you very much.

###########################

Major comments

############################

Comment 1: Need for improved texts and figures.

Response: We have revised the text and figures on the basis of your specific comments. Please see our responses to your specific comments.

Changes to the manuscript: Please see our responses to your specific comments.

#######

Comment 2: More information for reproducibility

Response: We have added more details about our methodology (e.g., software details and procedural details) along with the code that we used. We cannot attach the CMIP5 data because of their terms of use (http://cmip-pcmdi.llnl.gov/cmip5/terms.html), but we have attached the other data (data for observational databases) in the Supplement.

Changes to the manuscript: Please see our responses to your specific comments 6. Additionally, please see the Supplement.

#######

Comment 3: "The authors need to expand on the genuinely interesting parts, and offer more interesting, novel insights on how their results apply, and will be useful, to future work. The lack of reproducibility (above) means that it's also not clear how any of this would inform or be useful for modelers seeking to improve their software and science."

Response: This type of the manuscript was "Methods for assessment of models" (please see http://www.geoscientific-model-development.net/about/manuscript_types.html), and the main purpose of the paper was to demonstrate the possibility of using a machine learning algorithm to compare model outputs with observational databases; it was not aimed at identifying new mechanisms of the soil carbon cycle. We have improved the reproducibility of our analyses by adding more information about the methods along with the code and
applied data in the Supplement. We have modified the discussion to clearly convey our message.

Changes to the manuscript: Please see our responses to your specific comments, and Discussion in the revised manuscript.

##########################

Specific comments

##########################

Comment 1. Page 1, line 1: I'd suggest either "Data-mining analysis of the global..." or "Factors affecting the global..."

Response: We have changed the title to "Data-mining analysis of the global..."

Changes to the manuscript: "Data-mining analysis of the global distribution of soil carbon in observational databases and Earth system models"

#######

Comment 2. P. 1, lines 9, 20, and 26-27: these three short sentences could be deleted with no real loss

Response: We have deleted the first two sentences, but we prefer to retain the third sentence (lines 26-27) because, as mentioned above, we would like to emphasize this point.

Changes to the manuscript: We have deleted the first two sentences.

#######

Comment 3. P. 1, l. 25: "elucidate the nature" of the databases? Confusing

Response: We have rewritten the sentence.

Changes to the manuscript: (Page 1, line 24–25) "The results of this study should aid

in identifying the causes of mismatches between observational SOC databases and ESM outputs and improve the modelling of terrestrial carbon dynamics in ESMs."

#######

Comment 4. P. 2, l. 4: what recent study?

Response: This is Todd-Brown et al. 2013. We have added "(Todd-Brown et al., 2013)".

Changes to the manuscript: (Page 2, line7– 8) "a recent study (Todd-Brown et al., 2013) has found that. . ."

#######

Comment 5. 5. P. 3, l. 8-9: divided over what spatial scale? Some more detail in this entire para- graph would be useful

Response: A resolution was added.

Changes to the manuscript: (Page 3, line 30– 31) "The wetland ratio was calculated by dividing the number of wetland grids at 30 seconds by the total grids at $1°$."

#######

Comment 6. Methods: need to give version numbers CDO, R, and all packages used. Also, I'm shocked at the complete lack of any mention of data or code availability (no, that one sentence on p. 7 doesn't count). It's 2016, and I expect all code and data (at least that backing the main results) to be included as Supplementary info, or posted in a repository. It's not acceptable to produce results from a black box; see also http://www.geoscientific-model-development.net/about/code_and_data_policy.html

Response: We agree with the importance of openness. We have added details about the software used in this study and about the main code that we used. We have also attached the data used in this study. We cannot attach the CMIP5 output (please see the terms of use of CMIP5: http://cmip-pcmdi.llnl.gov/cmip5/terms.html), but our results

are easily reproduced by pasting CMIP5 data into the attached data and applying the data codes.

Changes to the manuscript: (Page 3, line 38–Page 4, line 2) "All global databases, except for the databases with a spatial resolution of 1° by default, including observational and ESM model outputs, were regridded to a spatial resolution of 1° for the analyses. Regridding of data in the NetCDF format was performed using the Climate Data Operators (CDO) software, version 1.6.9, provided by the Max Plank Institute for Meteorology (https://code.zmaw.de/projects/cdo). A bilinear interpolation, which is one of the most widely used algorithms, was used (remapbil in CDO)."

(Page 4, line 15– 23) "We used the open-source BRT package (brt.functions.R) in R software version 3.2.1 and 3.2.2 (R Core team, 2013) developed by Elith et al. (2008). The gbm package was used (version 2.1.1) to run the BRT package. The calculations were performed in Mac OS X (version 10.9.5 and version 10.10.5). To do so, the "windows" function in the "brt.functions.R" needed to be replaced with the "quartz" function in R. In practice, three parameters in the BRT package— the learning rate (lr), tree complexity (tc), and bag fraction (bg)—control the BRT performance. The lr determines the contribution of each tree, the tc controls the number of splits, and the bg is the proportion of data selected at each step. The number of trees was determined using the cross-validation method in the R package. The maximum number of trees was set to 15,000. The tc value was set to 5. We tested different lr (0.001, 0.005, 0.01, 0.05, 0.1) and bg values (0.5, 0.6, 0.7) and used the best parameter set for each database, but the changes in parameter values had little effect on the model performance."

Please see the Supplement.

#######

Comment 7. P. 4, l. 17: "Relationships with a mean annual temperature were relatively close to each other" – what does this mean? Clarify

Response: We have rephrased the sentence.

Changes to the manuscript: (Page 5, line 1) "Relationships with the mean annual temperature were similar."

#######

Comment 8. P. 4, l. 34: "The contribution of each variable varied between ESMs" ?

Response: We have rephrased the sentence.

Changes to the manuscript: (Page 5, line 19) "The contributions of some variables varied among ESMs"

#######

Comment 9. P. 4, l. 36: "large inconsistencies. . .demonstrated low contributions" – what?

Response: We have modified the sentence.

Changes to the manuscript: (Page 5, line 20–22) "Large inconsistencies between the observational databases and ESMs were found in the low contributions of clay content and the CN ratio and in the high contributions of NPP in ESMs (Figs. 5a and 5b)"

#######

Comment 10. P. 5, l. 23-24: this is an interesting point, and should be expanded upon. What are the implications, if the seemingly wide variety of CMIP5 models in fact uses a much smaller number of fundamental assumptions or modeling approaches? I'm pretty sure that Kathe Todd-Brown made this point in one of her papers; see also Alexander et al. (2015), 10.5194/gmd-8-1221-2015

Response: We have expanded this part.

Changes to the manuscript: (Page 6, line 4–21) "Analyses of the ESM outputs showed large variability, but the influential factors were predominantly similar among the ESMs

(Fig. 5). This similarity most probably indicates that the structures of the models that describe SOC dynamics in the ESMs are similar. One reason for the similarity is probably because some ESMs share common code (Alexander and Easterbrook, 2015). Another reason may be rooted in the basic structure of the soil carbon model: SOC is calculated as the balance between dead organic matter input to soil and carbon emissions from the decomposition of organic matter in soil, and these processes are influenced by temperature and water conditions. The SOC pool is characterized by its turnover time (decomposition constant). In general, decomposition exhibits an exponential response to temperature, which is more severe than its response to water. As a result, modelled SOC is strongly influenced by NPP (litter input), temperature, and turnover time, which have been demonstrated by previous studies (Exbrayat et al., 2014; Todd-Brown et al., 2013) and were also confirmed in our analyses. As shown in Table 2, SOC submodels in ESMs differ in the number of SOC pools and function types of temperature and moisture. Todd-Brown et al. (2013) have reported the absence of any pattern of agreement between ESM outputs and observational SOC databases with soil carbon pools, temperature and moisture sensitivity functions, and Exbrayat et al. (2014) have found that turnover times of SOC in ESM outputs are not affected by the number of SOC pools. Our analyses also indicated that a match or mismatch of major contributing factor between ESM outputs and observational databases are not strongly related to these properties of SOC submodels. Thus, it is likely that the spatial pattern of SOC from ESMS are more strongly affected by the basic structure, driving variables (NPP and temperature) and parameterisations (turnover time and influential parameters of temperature and moisture sensitivity) than by the number of pools and the function types of temperature and moisture sensitivity."

#######

Comment 11. P. 6, l. 13-14: "The use of temperature sensitivity. . ." - ?

Response: We have deleted the sentence.

Changes to the manuscript: Deleted.

#######

Comment 12. P. 7, l. 2-3: would such model-data fusion ever be possible, given the extremely long running time of modern ESMs?

Response: I agree. As you point out, model-data fusion of the whole ESM system is very difficult because of the long running time. In practice, I think that applying model-data fusion to a limited part of ESMs (e.g., ecosystem carbon cycle models) would be realistic. We have added the above points to the main text.

Changes to the manuscript: (Page 8, line 4– 7) "Constraining model parameters with observational databases through data assimilation, such as a Bayesian approach, would improve the performance of ESMs. Applying such model-data fusion to whole ESMs, however, would require a very long running time; therefore, model-data fusion to a part of an ESM (e.g., ecosystem carbon cycle model) would be realistic."

#######

Comment 13. Table A1: an URL or reference for each model would be useful

Response: We have added a URL for each model.

Changes to the manuscript: Please see Table 2 in the revised manuscript.

#######

Comment 14. Table A2: this classification was applied to. . .? Where is it from?

Response: We used soil texture data in the ISLSCPII database (Table 1), and a classification in the database was used (Table A2). Because the contribution of soil texture was not high, the relationships between the SOC and soil texture are not shown. To clarify that this soil texture classification is for the soil texture shown in Table 1, we have modified the caption of Table A1 and A2 in the revised manuscript.

Changes to the manuscript: (Page 15, Table A1) "Classification of soil texture in ISLSCPII (see Table 1)."

#######

Comment 15. Figures 1 and 2: these are so tiny I'm not sure they convey any information, really

Response: The maps have been redrawn.

Changes to the manuscript: Please see Fig. 1 and Fig. 2 in the revised manuscript.

#######

Comment 16. Figure 6 should be the central, most important figure of the entire paper–showing how variable importance compares between observational databases and ESMs–but it's very difficult to see what's going on. I'd suggest re-thinking this, and carefully con- sidering the most effective way to show this

Response: We have redrawn Fig. 6 using a box plot, and we have added a new figure to clearly show the results from each ESM. We have redrawn other figures, too.

Changes to the manuscript: Please see Fig. 5 in the revised manuscript.

Please also note the supplement to this comment:
http://www.geosci-model-dev-discuss.net/gmd-2016-138/gmd-2016-138-AC1-supplement.zip
* * *
[Figure]

**Fig. 1.** New Fig. 6 (Fig. 5 in the revised manuscript)

---

## Author Comment (AC2) · 7 Oct 2016

Dear Dr. Todd-Brown,

We greatly appreciate your constructive comments and suggestions. We have revised the manuscript on the basis of your comments, and the responses to the Major and Specific comments are found below. According to the editorial instructions, our response is structured as follows: (1) comments from Referees; (2) author's response; and (3) author's changes to the manuscript. Thank you very much.

#########################

General comments:

########################

Comment 1: Need for considering model structures

Response: We have added information about model structure to Table 2 in the revised manuscript. As reported in previous studies, we did not see clear influences of model structure on our results. We have also added a discussion.

Changes to the manuscript: (Page 6, line 4– 21) "Analyses of the ESM outputs showed large variability, but the influential factors were predominantly similar among the ESMs (Fig. 5). This similarity most probably indicates that the structures of the models that describe SOC dynamics in the ESMs are similar. One reason for the similarity is probably because some ESMs share common code (Alexander and Easterbrook, 2015). Another reason may be rooted in the basic structure of the soil carbon model: SOC is calculated as the balance between dead organic matter input to soil and carbon emissions from the decomposition of organic matter in soil, and these processes are influenced by temperature and water conditions. The SOC pool is characterized by its turnover time (decomposition constant). In general, decomposition exhibits an exponential response to temperature, which is more severe than its response to water. As a result, modelled SOC is strongly influenced by NPP (litter input), temperature, and turnover time, which have been demonstrated by previous studies (Exbrayat et al., 2014; Todd-Brown et al., 2013) and were also confirmed in our analyses. As shown in Table 2, SOC submodels in ESMs differ in the number of SOC pools and function types of temperature and moisture. Todd-Brown et al. (2013) have reported the absence of any pattern of agreement between ESM outputs and observational SOC databases with soil carbon pools, temperature and moisture sensitivity functions, and Exbrayat et al. (2014) have found that turnover times of SOC in ESM outputs are not affected by the number of SOC pools. Our analyses also indicated that a match or mismatch of major contributing factor between ESM outputs and observational databases are not strongly related to these properties of SOC submodels. Thus, it is likely that the spatial pattern of SOC from ESMS are more strongly affected by the basic structure, driving

variables (NPP and temperature) and parameterisations (turnover time and influential parameters of temperature and moisture sensitivity) than by the number of pools and the function types of temperature and moisture sensitivity."

Please see Table 2 in the revised manuscript.

######

Comment 2: Lack of other studies included in detail.

Response: We have added more detail regarding citation of other studies. Please see the specific response to comment 4.

Changes to the manuscript: (Page 2, line 14– 21) "Todd-Brown et al. (2013) have analysed soil carbon outputs from 11 ESMs from the fifth phase of the Coupled Model Intercomparison Project (CMIP5) and data from HWSD, and have found that the spatial variation of SOC from ESMs can be explained by net primary productivity (NPP) and temperature but that the spatial variation in HWSD cannot be explained by NPP and temperature. They have also found that the differences in SOC from ESMs are driven by differences in the simulated NPP and the parameterisation of soil heterotrophic respiration, not by differences in soil model structure in ESMs. The important influence of parameterisation of soil heterotrophic respiration (e.g., turnover time) on SOC in CMIP5 ESMs has also been suggested by Exbrayat et al. (2013)." (Page 6, line 9– 16) "The SOC pool is characterized by its turnover time (decomposition constant). In general, decomposition exhibits an exponential response to temperature, which is more severe than its response to water. As a result, modelled SOC is strongly influenced by NPP (litter input), temperature, and turnover time, which have been demonstrated by previous studies (Exbrayat et al., 2014; Todd-Brown et al., 2013) and were also confirmed in our analyses. As shown in Table 2, SOC submodels in ESMs differ in the number of SOC pools and function types of temperature and moisture. Todd-Brown et al. (2013) have reported the absence of any pattern of agreement between ESM outputs and observational SOC databases with soil carbon pools, temperature and

moisture sensitivity functions, and Exbrayat et al. (2014) have found that turnover times of SOC in ESM outputs are not affected by the number of SOC pools."

##########################

Specific comments

##########################

Comment 1: Please make it clearer that the ESMs are regressed against data products not other ESM output. While the modern ESM NPP and temperature distributions match better with current data products. There are some notable differences in modeled NPP in particular in the CMIP5 models and this could be a source of bias in the analysis.

Response: In this study, we downloaded the SOC output of ESMs from CMIP5 and examined the relationships between many variables in various data products listed in Table 1. We have revised the sentences to make this clearer. Furthermore, we have added the code to the manuscript Supplement, which should be helpful for understanding what we did. We have stated the cause for the variation in global SOC as the variation of modelled NPP.

Changes to the manuscript: (Page 2, line 28– 30) "We combined the potentially influential variables from many data products and SOC data from both observational databases with those by ESMs, and we examined the factors influencing the distribution of SOC and the relationships between these factors and SOC stocks.", (Page 7, line 20– 21) "Todd-Brown et al. (2013) have found that one of the major causes of variations in SOC among ESMs is differences in simulated NPP and that the strong control by NPP is not present in HWSD."

#######

Comment 2: P1L23-24 C:N ratio and clay content are in most ESMs in the allocation scheme. While it is intractable to investigate each modeling code directly, much of the

documentation for these ESMs includes Lignin:N ratio (similar to C:N ratios) and clay content mediating decomposition. CENTURY Parton et al 1988 use Lignin:N ratio and clay content for allocation parameters (IPSL-CM5 Krinner etal 2005 cite CENTURY: Parton et al 1988)

Response: As you noted, it is intractable to investigate each modelling code directly, and hence we believe that collecting and investigating each model code here is beyond the scope of this study. We have modified the sentences in the abstract and discussion.

However, we fully agree with the importance of investigating each model code, and we are very curious about how many processes in site-scale process-based models are incorporated in land ecosystem models in ESMs. For instance, the CENTURY model simulates C and N, and the dynamics of C and N influence each other. Hence, the CENTURY model SOC submodels without N do not fully capture the processes in the CENTURY model (for example, please see Figure S12 in Åďupek et al. 2016 Biogeosciences). These detailed comparisons should be required to identify the source of variation of SOC dynamics by ESMs in the future. To do so, full descriptions of the model structure and parameters are needed.

Changes to the manuscript: (Page 6, line 28– 35) "The SOC increased with increasing CN ratio in the observational databases (Fig. 4c), whereas the outputs of the ESM were insensitive to the CN ratio. Our results support the importance of properly incorporating the N cycle (e.g., control over decomposition, soil fertility, nutrient availability, and plant litter quality) into SOC models (Berg et al., 2001; Cotrufo et al., 2013; Fernández-Martínez et al., 2014; Liski et al., 2005; Tuomi et al., 2009; Åďupek et al., 2016). All of the ESMs, except for the CESM1 and NorESM in CMIP5, do not include terrestrial nitrogen processes (Todd-Brown et al., 2013). Including the nitrogen process has been suggested as an important improvement for the next model intercomparison (CMIP6) (Hajima et al., 2014; Zaehle et al., 2015). The results derived from our analysis support the importance of the appropriate inclusion of the N cycle in ESM models." (Page 6, line 36– Page 7, line 2) "Clay content is also often used as a regulator of the

decomposability of organic matter in the soil (e.g., CENTURY and RothC). Generally, high clay content inhibits organic matter decomposition in the soil. Furthermore, high clay content often results in low drainage and anaerobic soil conditions, which also inhibit organic matter decomposition. For IGBP-DIS, the clay content had as high a contribution as the CN ratio. The control of decomposability by clay content has been previously incorporated in site-scale process-based models (Parton et al., 1987) and may be incorporated in some ESMs, because soil carbon submodels in some ESMs are based on the CENTURY model (see the soil model history reported in Todd-Brown et al., 2014). However, regardless of incorporation of the control in decomposability by clay, our results suggest that the influence of clay on the carbon cycle is not well captured in present ESMs."

#######

Comment 3: P2L4-8 Should there be a citation here?

Response: This is Todd-Brown et al. 2013. We have added "(Todd-Brown et al., 2013)".

Changes to the manuscript: (Page 2, line7– 8) "a recent study (Todd-Brown et al., 2013) has found that. . ."

#######

Comment 4: P2 L8-11 A more in depth treatment of past attempts to disentangle drivers of data- model differences is called for here. Please expand on each of these treatments with particular attention to the ones that looked at the same models and data products the authors are using in this study. In addition, add something to the discussion to contrast your results with these studies.

Response: We have expanded the description of previous studies, added sentences and also omitted some sentences to contrast our results with previous work.

Changes to the manuscript: Please see the response to the General comment 1 and 2.

#######

Comment 5: Section 2.1 There needs to be some discussion about model structure in the ESM vs data products. These data products are typically constructed using correlation to the local environment (climate + land cover + geology) where the pedon was collected.

Response: We have included more sentences about the difference in the Introduction.

Changes to the manuscript: (Page 1, line 38– Page 2, line5) "These databases incorporate observed data points with global coverage, although there are biases in the spatial distribution or densities of the data points. In these databases, gridded SOC have been generated on the basis of inter-extrapolation of model outputs derived from analysis of observed SOC data points. Earth system models (ESMs), which have been developed to understand the current climate and to provide future climate projections, incorporate the terrestrial carbon cycle, including SOC. In ecosystem carbon cycle models of ESMs, SOC is calculated as the balance between dead organic matter input into soil and carbon emission from the decomposition of organic matter in soil, and these processes are influenced by temperature and water conditions. Compared with the observational estimation of SOC, the SOC distribution in ESMs involves more process-oriented simulations."

#######

Comment 6: Section 2.1: Please summarize the methods used for each specific data product. For ESMs a discussion of their sensitivity functions and pool structure is appropriate (note that BCC was incorrectly stated to have their N-cycle turned on for CMIP5 in Todd-Brown et al 2013).

Response: We summarized the methods used for observational SOC databases. "BCC" was omitted from the list of ESMs with an N cycle. We have added a discussion about the model structure.

[Figure]

Changes to the manuscript: (Page 2, line 35–Page 3, line 9) "We used SOC data from two global and one northern observational database. The first global database was the HWSD (FAO/IIASA/ISRIC/ISSCAS/JRC, 2012). The HWSD is a global database of soil physiochemical properties that has been developed by the International Institute for Applied Systems Analysis (IIASA) and the Food and Agriculture Organization of the United Nations (FAO) in collaboration with the International Soil Reference and Information Centre (ISRIC) -World Soil Information, the European Commission Joint Research Centre (JRC), and the Institute of Soil Science, Chinese Academy of Sciences (ISSCAS). The database was constructed by compiling the European Soil Database (ESDB), a 1:1 million soil map of China, various regional SOTER databases (SOTWIS Database), and a soil map of the world from the FAO. We used an SOC stock database obtained with HWSD from the Joint Research Centre (JRC) (Hiederer and Köchy, 2011) (Fig. 1a). The second database included global gridded surfaces of selected soil characteristics (IGBP-DIS) (Global Soil Data Task Group, 2000) (Fig. 1b), which contains gridded soil physiochemical properties. The database has been developed by the Global Soil Data Task Group of the international Geosphere Biosphere Programme's (IGBP) Data and Information System (DIS), and the database was generated by linking the pedon records in the Global Pedon Database to the FAO/UNESCO digital soil map of the world. The third database was the Northern Circumpolar Soil Carbon Database, version 2 (NCSCD) (Hugelius et al., 2013; Tarnocai et al., 2009) (Fig. 1c). This database is a spatial database of SOC stock of the northern circumpolar permafrost region. The soil map data were obtained from different regions/countries (e.g., USA, Canada, Russia etc.) and were harmonized. The NCSCD were based on 1778 pedon data points."

Please see the Table 2.

(Page 6, line 4– 21)"Analyses of the ESM outputs showed large variability, but the influential factors were predominantly similar among the ESMs (Fig. 5). This similarity most probably indicates that the structures of the models that describe SOC dynamics in the ESMs are similar. One reason for the similarity is probably because some ESMs share common code (Alexander and Easterbrook, 2015). Another reason may be rooted in the basic structure of the soil carbon model: SOC is calculated as the balance between dead organic matter input to soil and carbon emissions from the decomposition of organic matter in soil, and these processes are influenced by temperature and water conditions. The SOC pool is characterized by its turnover time (decomposition constant). In general, decomposition exhibits an exponential response to temperature, which is more severe than its response to water. As a result, modelled SOC is strongly influenced by NPP (litter input), temperature, and turnover time, which have been demonstrated by previous studies (Exbrayat et al., 2014; Todd-Brown et al., 2013) and were also confirmed in our analyses. As shown in Table 2, SOC sub-models in ESMs differ in the number of SOC pools and function types of temperature and moisture. Todd-Brown et al. (2013) have reported the absence of any pattern of agreement between ESM outputs and observational SOC databases with soil carbon pools, temperature and moisture sensitivity functions, and Exbrayat et al. (2014) have found that turnover times of SOC in ESM outputs are not affected by the number of SOC pools. Our analyses also indicated that a match or mismatch of major contributing factor between ESM outputs and observational databases are not strongly related to these properties of SOC submodels. Thus, it is likely that the spatial pattern of SOC from ESMS are more strongly affected by the basic structure, driving variables (NPP and temperature) and parameterisations (turnover time and influential parameters of temperature and moisture sensitivity) than by the number of pools and the function types of temperature and moisture sensitivity."

#######

Comment 7: P2 L33-35 Be more convincing about averaging models from the same center, there is some clustering analysis that is in the Supplemental of Todd-Brown et al 2013 that could support this.

Response: We have cited Todd-Brown et al. 2013, who have found that ESMs from

the same climate centre generate very similar distributions of SOC.

Changes to the manuscript: (Page 3, line 17–18) ": Todd-Brown et al. (2013) showed through a hierarchical cluster analysis that SOC distributions were very similar among ESMs from the same climate centre."

#######

Comment 8: P2 L34-35 Todd-Brown et al 2013 averaged all ensembles that were available at the time, this statement is incorrect. Please either provide a different justification for only considering one ensemble or, preferably, go back and re-analyze the data with the multi-ensemble mean (even better if you can incorporate the modeled uncertainty).

Response: We apologize for the incorrect description and have corrected it. We have cited other references that use only r1i1p1; most ESMs have this ensemble member output (Dirmeyer et al. Journal of Hydrometeorology 2013.; Chang et al. Journal of Geophysical Research 2012; Kumar et al. Climate Dynamics 2014; Jiang et al. Journal of Climate 2015).

Changes to the manuscript: (Page 3, line 19–21) "The notation "r1i1p1" is an identifier of the model simulation and is an ensemble member that is often used for analyses (Chang et al., 2012; Dirmeyer et al., 2013; Jiang et al., 2015; Kumar et al., 2014)."

#######

Comment 9: Section 2.4 What regridding algorithm did you use? There are several options in CDO, not all are appropriate for soil data, temperature and NPP. Please discussion which algorithm was used and why.

Response: We used "remapbil" (a bilinear interpolation) in CDO for the soil data. We used this algorithm simply because this is one of the most widely used algorithms for regridding. This study focuses only on the spatial pattern of SOC, and the total amounts of SOC were beyond the scope of this study. We believe that the difference in regridding algorithms would not affect the conclusions of this study, but we will willingly

conduct a recalculation if you strongly recommend a specific algorithm.

Changes to the manuscript: (Page 4, line 1– 2) "A bilinear interpolation, which is one of the most widely used algorithms, was used (remapbil in CDO)."

#######

Comment 10: P5 L1 Describe the results here in addition to referencing the figure.

Response: The description of the results was after the sentence P5 L1 in the former manuscript. We agree with your comment that this sentence was unnecessary. We have modified this paragraph by inserting figure numbers after descriptions.

Changes to the manuscript: (Page 5, line 24– 30) "The relationships between SOC and certain variables substantially varied among the ESM databases (Fig. 6a–e), particularly in the mean annual temperature (Fig. 6a). The SOC decreased with increasing mean annual temperature (Fig. 6a) but increased with increasing precipitation (Fig. 6b) and NPP (Fig. 6e). The mean of the relationship with mean annual temperature for ESMs was highly consistent with that in the HWSD and IGBP-DIS databases of the temperature range $-5$–$15\,^\circ$C (Fig. 6a). The increasing trend with increasing NPP in ESMs was consistent with that of the HWSD, particularly below approximately 500 g C m$-2$ of NPP (Fig. 6e). Although the wetland ratio did not contribute to the ESMs (Fig. 6a) with respect to land cover, permanent wetlands had higher SOC (Fig. 6d)."

#######

Comment 11: P7 L15 A BRT tutorial is not appropriate to cite under 'Code availability'. Please either link or reference as SI to the actual code used in this analysis (preferred) or remove this section.

Response: We have added the codes and data for the observational databases to the Supplement.

Changes to the manuscript: (Page 8, line 18– 20) "The R code, with a tutorial for BRT, is available in the supplementary material of Elith et al. (2008) (http://onlinelibrary.wiley.com/doi/10.1111/j.1365-2656.2008.01390.x/full). The codes and data for the observational databases are available in the Supplement."

Please see the Supplement.

Please also note the supplement to this comment: http://www.geosci-model-dev-discuss.net/gmd-2016-138/gmd-2016-138-AC2-supplement.zip
* * *
[Figure]

**Fig. 1.** New Fig. 6 (Fig. 5 in the revised manuscript)

---

## Author Response (AR2)

Dear Dr. Bond-Lamberty,

We would like to thank you for your remarks and comments on our previous revision. We have revised the manuscript again, and our responses to specific comments can be found below. Thank you very much.

###########################

Specific comments

###########################

Comment 1. Page 1, l. 22-24: this sentence is a bit unclear; reword if possible

Response: We have reworded the sentence.

Changes to the manuscript: (Page 1, line 21-23) "A comparison of the influential factors at a global scale revealed that the most distinct differences between the SOCs from the observational databases and ESMs were the low clay content and CN ratio contributions and the high NPP contribution in the ESMs."
#######

Comment 2. P. 1, l. 29: and DOC losses

Response: We have included the DOC losses.

#######

Comment 3. P. 1, l. 32-33: update with Crowther et al. (2016), just published in Nature

Response: We have updated the sentences with the citation for Crowther et al. (2016).

#######

Comment 4. P. 2, l. 11: start new paragraph at "Several studies…"

Response: We have modified this sentence.

#######

Comment 5. P. 2, l. 15, 16, 18: delete "have" (also applies to p. 6, l. 14, 16)

Response: We have deleted "have".

#######

Comment 6. P. 6, l. 22-25: could probably delete these sentences

Response: We have deleted these sentences.

########

Comment 7. P. 6, l. 31: "None of the ESMs…CMIP5, include"

Response: We have rewritten this sentence. (P. 6, l. 23)

########

Comment 8. P. 7, l. 29-31: could be removed

Response: We have deleted these sentences.

Dear Dr. Todd-Brown,

We would like to thank you for your remarks on our previous revision. We greatly appreciate your constructive comments and suggestions. We have revised the manuscript based on your comments, and our responses are provided below. According to the editorial instructions, our response is structured as follows: (1) comments from the Referees; (2) author's response; and (3) author's changes to the manuscript. Thank you very much.

#######################

General comments:

#######################

#######

Comment: First I would like to thank the authors for their revisions to the previous version. The manuscript is much improved and, in particular, the addition of the scripts lends critical reproducibility to the study. I would like to see more context for the CMIP5 analysis and soil carbon models in general, the description of soil carbon models simulating input/output fluxes as being the reason the ESMs fall out so similarly is particularly troubling (see below). I'm also very concerned with the treatment of the data product as observational truth instead of model output in and of itself (see below). Finally there are some additional readability issues that need to be smoothed out. Other issues brought up are relatively minor or mere suggestions.

Response: We appreciate your comments on our former manuscript. We have revised the manuscript according to your comments again, and details on the revisions are listed below.

#######################

Specific comments:

#######################

#######

Comment 1: At the risk of bringing up a new topic in a re-review, I would urge the authors to consider reframing their study results: both the data products and ESM simulations are, in fact, model results. In particular data products extrapolate empirical relationships observed between environmental variables and SOC using a paint-by-numbers or machine learning scheme, where as ESMs extrapolate relationships between input/output fluxes and SOC plus environmental variables. Both products are the result of modelling work but with separate assumptions. The value in comparing the two is

that the assumptions are independent and if they were to agree we would have an increased confidence in our overall understanding of the system However, it is debatable whether one or the other is closer to reality. The current framing holds up the data products as 'TRUTH" without acknowledging that there are significant uncertainties that go into creating these products.

Response: We agree that both the observational data products (e.g., HWSD) and the ESM outputs are derived by extrapolating/modelling using assumptions and that both results present significant uncertainty. In addition, we think that high field-scale variability is likely one of the important sources of this uncertainty. Although we did not mean to say that the observational databases are the truth, we do believe that at this stage, observational SOC datasets are relatively closer to reality than ESMs output because the observational SOC products are directly generated from SOC observations, while the reproducibility of observed SOC is not necessarily of the highest priority in ESM modelling in CMIP5. In fact, many previous studies that have attempted to improve SOC outputs from models use these observational SOC products to evaluate SOC outputs from models, and these studies indicate that observational SOC products are relatively closer to reality and benchmarks (Anav et al. 2013; Todd-Brown et al. 2013; Hararuk et al. 2014; Wieder et al. 2014; Tian et al. 2015 etc.). We admit that it is important to let the readers know that even observational databases include significant uncertainty. We have added a new paragraph in the Discussion and a sentence in the Concluding remarks and the Introduction regarding this uncertainty.

Changes to the manuscript:

(Page 2, line 10-12) "Compared with the SOC distribution derived from ESMs, SOC estimates derived from SOC observations are more data oriented; however, even the observational databases include significant uncertainty because of errors in the source data and building processes (Köchy et al., 2015; Todd-Brown et al., 2013)."

(Page 8, line 13-19) "Observational databases are directly generated from SOC observations; therefore, these databases should be closer to the real SOC distribution than databases based on ESM outputs. Hence, observational databases are often used as benchmarks to evaluate the outputs of ESMs. Still, we should be aware that these observational databases are generated using assumptions, certain algorithms, and uncertain inputs. In particular, the uncertainty in the SOC for the northern regions is high in the observational databases (Fig. 1), which was also observed in the results from our BRT analysis (Fig. 3 c–e). When the estimated SOC distribution from various approaches, such as data-driven and process-oriented modelling, are consistent, then our estimations have high confidence."

(Page 8, line 39-41) "Although observational estimations of SOC are still under development and have significant uncertainty, the consistency between observational

SOC database results and ESM outputs will enhance our confidence in predicting SOC dynamics under climate change."

#######

Comment 2: I would like to suggest, but not require, a more active title that alludes to the key conclusion of the manuscript.

Response: One of the main objectives of this study is addressing how a data-mining algorithm can be used to assess model outputs; thus, we prefer to retain this title.

#######

Comment 3: There are significant readability issues with the current version of this manuscript (introduction and discussion are rough, methods and results sections are ok). I would suggest that the authors consider reworking the phrasing and logical structure of the introduction and discussion sections. Example: p1 l 57-40 it is unclear what 'basis of inter-extrapolation of model outputs' means in this context.

Response: We have revised the Introduction and Discussion by rephrasing and restructuring the paragraphs, and these sections were checked by a professional English editing service again.

Changes to the manuscript: Please see the Introduction and Discussion.

#######

Comment 4: P2 L13-14 Microbial explicit models tend to produce unrealistic temporal oscillations (Wang et al., 2014) it's not clear they are more realistic at the global scale. However there are a number of general reviews that have suggested various mechanisms and processes that could be included in the next generation of soil decomposition models which could be reviewed in this paragraph including not limited to (Luo et al., 2015; Ostle et al., 2009; Wieder et al., 2015)

Response: We have deleted this sentence, inserted an improved sentence in the Discussion, and included these citations for these review papers.

Changes to the manuscript:

(Page 8, line 21-24) "For example, including microbial dynamics in SOC models may improve projections of global soil carbon by ESMs (Wieder et al., 2013), although

models that include these dynamics are still in development (see Wang et al., 2014, 2016; Wieder et al., 2015)."

(Page 6, line 35-37) "The potential mechanisms, parameterization, and other modelling issues for next-generation ESMs are not limited to those listed above and have been thoroughly discussed elsewhere (Luo et al., 2016; Ostle et al., 2009; Wieder et al., 2015). "

########

Comment 5: There have been several assessments made of the soil carbon dynamics in CMIP5 models beyond the studies by Todd-Brown et al and Exbrayat et al, including but not limited to (Anav et al., 2013; Arora et al., 2013; Friedlingstein et al., 2014; Koven et al., 2015; Shao et al., 2013). The authors need to include an overview of the general findings of this body of work.

Response: To our knowledge, few studies have focused on SOC distributions and stocks obtained by CMIP5, and the main topic of these papers was SOC (Todd-Brown et al and Exbrayat et al.). However, the suggested papers focus on SOC in a much wider context, and we think that these studies are also important. The suggested papers (Arora et al., 2013, Friedlingstein et al., 2014, Shao et al., 2013 and Koven et al., 2015, and particularly Anav et al. 2013) were also relevant for understanding the factors that influence SOC, and we included their general findings in the Introduction and Discussion.

Changes to the manuscript: Please see the Introduction.

########

Comment 6: The methods section and scripts are much improved. Thank you!

Response: Thank you for your feedback on the improvements to our previous manuscript.

########

Comment 7: P5 Why did the 'data' drivers differ so much? How were the data products constructed differently that might lead to this pattern?

Response: Influential variables are likely consistent on a global scale between the HWSD and IGBP-DIS, although they differed when we focused on the northern regions (Figure 3 (c-e)). This finding was likely because of the different base maps and different sources of pedon data. We have included this issue in the revised manuscript.

Changes to the manuscript: (Page 8, line 15-17) "Still, we should be aware that these observational databases are generated using assumptions, certain algorithms, and uncertain inputs. In particular, the uncertainty in the SOC for the northern regions is high in the observational databases (Fig. 1), which was also observed in the results from our BRT analysis (Fig. 3 c–e)."

#######

Comment 8: P6 L7-9 While this is entirely true, the ESM soil organic carbon is simulated by modelling the input/output fluxes, this does not account for the behavioural similarity seen in this study. One could model input and output fluxes as constants or as a chaotic dynamical system and this would generate completely divergent patterns. Their similarities likely lay in the fact they are all first order linear ordinary differential equations, all be it non-autonomous ODEs.

Response: We agree with this recommendation and have included this information in the paragraph.

Changes to the manuscript: (Page 8, line 9-11) "From a mathematical perspective, the similarity is likely fundamentally based on the description of these SOC dynamics by a series of first-order linear ordinary differential equations that are not autonomous (Manzoni and Porporato, 2009; Sierra and Müller, 2015). With these equations, the outputs generally do not show chaotic behaviours."

#######

Comment 9: P6 L14-18 I like the discussion points about the lack of structure driven differences between the ESMs.

Response: Thank you.

#######

Comment 10: P7 L6 These Q10 values from Todd-Brown 2013, 2014 were inferred Q10 values, not documented Q10. They were fitted during the post-hoc however the sentence currently reads as if they are documented Q10. Please clarify.

Response: We apologize for the poor description, which has been clarified in the revised manuscript.

Changes to the manuscript: (Page 6, line 37-39) "Based on an analysis of the output of heterotrophic respiration, the temperature sensitivity (e.g., $Q_{10}$ value) of soil organic matter decomposition in the ESMs has been reported as 1.4 to 2.2 (Todd-Brown et al., 2014)."

#######

Comment 11: P7 L9-11 This seems a bit disconnected, not sure what you are trying to say here.

Response: We were attempting to indicate that our analysis did not provide a parameter for the temperature sensitivity of plant productivity and decomposition that can be used in the ESMs. We have inserted the word "parameter" to clarify the meaning of the sentence.

Changes to the manuscript: (Page 7, line3-5) "The relationships between the SOC and temperature obtained in this study include the integration of the temperature sensitivity of both plant production and soil organic decomposition and thus do not provide the temperature sensitivity parameter of individual processes for ESMs."

#######

Comment 12: P8 L5-7 This is a contradictory sentence.

Response: We have rewritten these sentences.

Changes to the manuscript: (Page 8, line 31-34) "Although its application to a part of an ESM (e.g., ecosystem carbon cycle model) is realistic in consideration of the long running time, constraining model parameters with observational databases via data assimilation, such as a Bayesian approach, would improve the performance of ESMs."

#######

Comment 13: P8 L18-20 I believe that the authors mean to say something like. "The R code, with tutorial, for the BRT algorithm is …" that being said this is more appropriate as a cited reference placed in the methods section then part of the code availability statement.

Response: We have modified the statement for the code availability and included the availability in the methods section as well.

Changes to the manuscript:

(Page 4, line 25) "The R code for the BRT algorithm is available in the supplementary material of Elith et al. (2008)."

(Page 9, line 10-12) "The R code, with a tutorial, for the BRT algorithm is available in the supplementary material of Elith et al. (2008) (http://onlinelibrary.wiley.com/doi/10.1111/j.1365-2656.2008.01390.x/full). The codes and data for the observational databases are available in the Supplement."

*#######*

Comment 14: Table 2: According to CMIP5 data use this needs a very specific format for the model group names and other material. Currently it does not conform.

Response: For ensembles, we added the official model names in the footnote of Table 2. We have changed "INMCM4" to "INM-CM4" to follow the official model names.

Changes to the manuscript: Please see Table 2 and Figure 2 and 5.

*#######*

Comment 15: The index id's don't really add anything to the model names, consider removing this from Table 2 and Figure 5.

Response: We have deleted the ids from Table 2 and Figure 5.

Changes to the manuscript: Please see Table 2 and Figure 5.

[revised manuscript text omitted]